# LEARNING GFLOWNETS FROM PARTIAL EPISODES FOR IMPROVED CONVERGENCE AND STABILITY

## ABSTRACT

Generative flow networks (GFlowNets) are a family of algorithms for training a sequential sampler of discrete objects under an unnormalized target density and have been successfully used for various probabilistic modeling tasks. Existing training objectives for GFlowNets are either local to states or transitions, or propagate a reward signal over an entire sampling trajectory. We argue that these alternatives represent opposite ends of a gradient bias-variance tradeoff and propose a way to exploit this tradeoff to mitigate its harmful effects. Inspired by the TD($\lambda$) algorithm in reinforcement learning, we introduce *subtrajectory balance* or SubTB($\lambda$), a GFlowNet training objective that can learn from partial action subsequences of varying lengths. We show that SubTB($\lambda$) accelerates sampler convergence in previously studied and new environments and enables training GFlowNets in environments with longer action sequences and sparser reward landscapes than what was possible before. We also perform a comparative analysis of stochastic gradient dynamics, shedding light on the bias-variance tradeoff in GFlowNet training and the advantages of subtrajectory balance.

## 1 INTRODUCTION

Generative flow networks (GFlowNets; Bengio et al., 2021a) are generative models that construct objects lying in a target space $\mathcal{X}$ by taking sequences of actions sampled from a learned policy. GFlowNets are trained so as to make the probability of sampling an object $x \in \mathcal{X}$ proportional to a given nonnegative reward $R(x)$. GFlowNets' use of a parametric policy that can generalize to states not seen during training makes them a competitive alternative to methods based on local exploration in various probabilistic modeling tasks (Bengio et al., 2021a; Malkin et al., 2022; Zhang et al., 2022; Jain et al., 2022; Deleu et al., 2022).

GFlowNets solve the variational inference problem of approximating a target distribution over $\mathcal{X}$ with the distribution induced by the sampling policy, and they are trained by algorithms reminiscent of reinforcement learning (although GFlowNets model the diversity present in the reward distribution, rather than maximizing reward by seeking its mode). In most past works (Bengio et al., 2021a; Malkin et al., 2022; Zhang et al., 2022; Jain et al., 2022), GFlowNets are trained by exploratory sampling from the policy and receive their training signal from the reward of the sampled object. The flow matching (FM) and detailed balance (DB) learning objectives for GFlowNets proposed in Bengio et al. (2021a;b) resemble temporal difference learning (Sutton & Barto, 2018).

A third objective, trajectory balance (TB), was proposed in Malkin et al. (2022) to address the problem of slow temporal credit assignment with the FM and DB objectives. The TB objective propagates learning signals over entire episodes, while the temporal difference-like objectives (FM and DB) make updates local to states or actions. It has been hypothesized by Malkin et al. (2022) that the improved credit assignment with TB comes at the cost of higher gradient variance, analogous to the bias-variance tradeoff seen in temporal difference learning (TD($n$) or TD($\lambda$)) with different eligibility trace schemes (Sutton & Barto, 2018; Kearns & Singh, 2000; van Hasselt et al., 2018; Bengio et al., 2020). This hypothesis is one of the starting points for the present paper.

In this paper, we propose a new learning objective for GFlowNets, called *subtrajectory balance* (SubTB, or SubTB($\lambda$) when its real-valued hyperparameter $\lambda$ is specified). Building upon theoretical results of Bengio et al. (2021b); Malkin et al. (2022), we show how the SubTB($\lambda$) objective allows the flexibility of learning from partial experiences of any length. Experiments on two synthetic and four real-world domains support the following empirical claims:

(1) SubTB($\lambda$) improves convergence of GFlowNets in previously studied environments: models trained with SubTB($\lambda$) approach the target distribution in fewer training iterations and are less sensitive to hyperparameter choices.

(2) SubTB($\lambda$) enables training of GFlowNets in environments where past approaches perform poorly due to sparsity of the reward function or length of action sequences.

(3) The benefits of SubTB($\lambda$) are explained by lower variance of the stochastic gradient, with the parameter $\lambda$ allowing interpolation between the high-bias, low-variance DB objective and the low-bias, high-variance TB objective.

## 2 METHOD

### 2.1 PRELIMINARIES

In this section, we summarize the necessary preliminaries on GFlowNets. We follow the notation of Malkin et al. (2022), to which the reader is directed for a more thorough exposition written with a view towards motivating the trajectory and subtrajectory balance objectives. A deeper introduction is given in Bengio et al. (2021b).

Let $G = (\mathcal{S}, \mathcal{A})$ be a directed acyclic graph. The vertices $s \in \mathcal{S}$ are called *states* and the directed edges $(u \to v) \in \mathcal{A}$ are *actions*. If $(u \to v)$ is an edge, we say $v$ is a *child* of $u$ and $u$ is a *parent* of $v$. There is a unique *initial state* $s_0 \in \mathcal{S}$ with no parents. States with no children are called *terminal*, and the set of terminal states is denoted by $\mathcal{X}$.

A *trajectory* or an *action sequence* is a sequence of states $\tau = (s_m \to s_{m+1} \to \ldots \to s_n)$, where each $(s_i \to s_{i+1})$ is an action. The trajectory is *complete* if $s_m = s_0$ and $s_n$ is terminal. The set of complete trajectories is denoted by $\mathcal{T}$.

A *(forward) policy* is a collection of distributions $P_F(-|s)$ over the children of every nonterminal state $s \in \mathcal{S}$. A forward policy determines a distribution over $\mathcal{T}$ by

$$P_F(\tau = (s_0 \to \ldots \to s_n)) = \prod_{i=0}^{n-1} P_F(s_{i+1}|s_i). \tag{1}$$

Any distribution over complete trajectories that arises from a forward policy satisfies a Markov property: the marginal choice of action out of a state $s$ is independent of how $s$ was reached. Conversely, any Markovian distribution over $\mathcal{T}$ arises from a forward policy (Bengio et al., 2021b).

A forward policy can thus be used to sample terminal states $x \in \mathcal{X}$ by starting at $s_0$ and iteratively sampling actions from $P_F$, or, equivalently, taking the terminating state of a complete trajectory $\tau \sim P_F(\tau)$. The marginal likelihood of sampling $x \in \mathcal{X}$ is the sum of likelihoods of all complete trajectories that terminate at $x$.

Suppose that a nontrivial (not identically 0) nonnegative reward function $R : \mathcal{X} \to \mathbb{R}_{\geq 0}$ is given. The learning problem solved by GFlowNets is to estimate a policy $P_F$ such that the likelihood of sampling $x \in \mathcal{X}$ is proportional to $R(x)$. That is, there should exist a constant $Z$ such that

$$R(x) = Z \sum_{\tau = (s_0 \to \ldots \to s_n = x)} P_F(\tau) \quad \forall x \in \mathcal{X}. \tag{2}$$

If (2) is satisfied, then $Z = \sum_{x \in \mathcal{X}} R(x)$.

### 2.2 GFLOWNET TRAINING OBJECTIVES

Because the sum in (2) may be intractable to compute, it is in general not possible to directly convert this constraint into a training objective. To solve this problem, GFlowNet training objectives introduce auxiliary variables in the parametrization in various ways, but all have the property that (2) is satisfied at the global optimum. The key properties of these objectives are summarized in Table 1.

**Flow matching (FM; Bengio et al., 2021a).** Motivating the 'flow network' terminology, Bengio et al. (2021a) proved that (2) is satisfied if $P_F$ arises from an *edge flow function* satisfying certain constraints. Namely, an assignment $F : \mathcal{A} \to \mathbb{R}_{\geq 0}$ of a nonnegative number (flow) to each action defines a policy via

$$P_F(t|s) = \frac{F(s \to t)}{\sum_{t':(s \to t') \in \mathcal{A}} F(s \to t')}. \tag{3}$$

A sufficient condition for the terminating distribution of $P_F$ to be proportional to the reward $R(x)$ is that a family of flow-matching (flow in = flow out) conditions is satisfied at all interior states and a

Table 1: Summary of GFlowNet training objectives.

| Objective | Parametrization | Locality |
|---|---|---|
| Flow matching | edge flow $F(s{\to}t; \theta)$ | state $s$ |
| Detailed balance | state flow $F(s; \theta)$, policies $P_F(-\|-; \theta)$, $P_B(-\|-; \theta)$ | action $s{\to}t$ |
| Trajectory balance | initial state flow $Z_\theta$, policies $P_F(-\|-; \theta)$, $P_B(-\|-; \theta)$ | complete trajectory $\tau$ |
| Subtrajectory balance | state flow $F(s; \theta)$, policies $P_F(-\|-; \theta)$, $P_B(-\|-; \theta)$ | (partial) trajectory $\tau$ |

family of reward-matching conditions is satisfied at terminal states:

$$\sum_{s:(s\to t)\in\mathcal{A}} F(s{\to}t) = \sum_{u:(t\to u)\in\mathcal{A}} F(t{\to}u) \qquad \forall t \in \mathcal{S} \setminus (\mathcal{X} \cup \{s_0\}),$$

$$\sum_{s:(s\to x)\in\mathcal{A}} F(s{\to}x) = R(x) \qquad \forall x \in \mathcal{X}. \qquad (4)$$

The flow $F(s{\to}t)$ is then proportional to the marginal likelihood that a complete trajectory sampled from $P_F$ includes the action $s{\to}t$.

In Bengio et al. (2021a), a GFlowNet is described by a parametric estimate of the edge flow function, $F(u{\to}v; \theta)$ (a neural net with parameters $\theta$). These conditions can be converted into objectives that are minimized when (4) is satisfied. For example, the flow-matching objective at a nonterminal state $s$ is defined by

$$\mathcal{L}_{\text{FM}}(s) = \left(\log \frac{\sum_{s:(s\to t)\in\mathcal{A}} F(s{\to}t; \theta) + \epsilon}{\sum_{u:(t\to u)\in\mathcal{A}} F(t{\to}u; \theta) + \epsilon}\right)^2, \qquad (5)$$

where $\epsilon$ is a smoothing constant that can safely be set to 0 if the flows are constrained to be strictly positive, and a similar objective (or a constraint by construction) is defined to force the flow $F(s{\to}x)$ into terminal states $x$ to match $R(x)$. If these objectives are globally minimized for all states $s$, then the policy $P_F(-\|-; \theta)$ defined by $F(-; \theta)$ via (3) satisfies (2), with $Z = \sum_{t:(s_0\to t)\in\mathcal{A}} F(s{\to}t; \theta) = \sum_{x\in\mathcal{X}} R(x)$. The question of how to sample states $s$ for training is discussed below.

**Detailed balance (DB; Bengio et al., 2021b; Malkin et al., 2022).** In the DB parametrization, a forward policy model $P_F(-\|-; \theta)$ is learned directly, jointly with two additional objects: a *backward policy* model $P_B(-\|-; \theta)$, which can predict a distribution over the parents of any noninitial state, and a *state flow* function $F(s; \theta)$ (typically parametrized in the log domain). The detailed balance conditions state that

$$F(s; \theta)P_F(t|s; \theta) = F(t; \theta)P_B(s|t; \theta) \qquad (6)$$

for all actions $(s{\to}t)$ and $F(x; \theta) = R(x)$ for $x$ terminal. Satisfaction of these conditions for all actions $(s{\to}t)$ and $x \in \mathcal{X}$ implies that $P_F$ samples proportionally to the reward (i.e., satisfies (2), with $Z = F(s_0)$). The DB condition (6) can be converted into a squared log-ratio objective $\mathcal{L}_{\text{DB}}(s{\to}t)$ in the same way that (4) yields (5), and $\mathcal{L}_{\text{DB}}(s{\to}t)$ can be optimized over sampled actions $(s{\to}t)$.

**Trajectory balance (TB; Malkin et al., 2022).** The parametrization required for the TB objective includes forward and backward policy models $P_F(-\|-; \theta)$ and $P_B(-\|-; \theta)$, as well as an estimate $Z_\theta$ of the constant of proportionality in (2). Satisfaction of the following condition for all complete trajectories $\tau = (s_0{\to}\ldots{\to}s_n)$ implies that (2) is satisfied:

$$Z_\theta P_F(\tau; \theta) = R(s_n)P_B(\tau|s_n; \theta), \qquad (7)$$

where we have used the conventions

$$P_F(\tau; \theta) = \prod_{i=0}^{n-1} P_F(s_{i+1}|s_i; \theta), \quad P_B(\tau|s_n; \theta) = \prod_{i=0}^{n-1} P_B(s_i|s_{i+1}; \theta).$$

The condition (7) can again be made into a squared log-ratio objective $\mathcal{L}_{\text{TB}}(\tau)$ and optimized for complete trajectories $\tau$ taken from some training policy. In Malkin et al. (2022), the TB objective was empirically demonstrated to have better convergence properties than FM and DB on various problem domains.

**Training policy and exploration.** Global minimization of the FM, DB, and TB objectives for all values of their respective arguments (states, actions, or complete trajectories) implies satisfaction of (2). Therefore, given a sufficiently expressive model and convergence of the optimization procedure, a GFlowNet policy that samples $x$ with likelihood proportional to $R(x)$ can be trained by minimizing any of these losses over a distribution with full support, enabling offline training of GFlowNets. As

in other RL algorithms, the distribution over sampled states, actions, or episodes can be fixed and off-policy, or can vary over the course of training and use available information about terminal states in interesting ways (Zhang et al., 2022; Deleu et al., 2022). The simplest approach, which is also taken in this paper, is on-policy learning or a very similar off-policy variant that flattens the current policy to ensure exploration. Complete trajectories $\tau = (s_0 \to \ldots \to s_n)$ are sampled from the forward policy $P_F(-|-;\theta)$ (tempered or mixed with a uniform policy with a small weight so as to ensure full support and exploration). One then takes gradient descent steps on $\mathcal{L}_{\text{TB}}(\tau)$, on $\mathcal{L}_{\text{DB}}(s_i \to s_{i+1})$ over all actions in $\tau$, or on $\mathcal{L}_{\text{FM}}(s_i)$ for all intermediate states in $\tau$.

The GFlowNets in this paper are trained on-policy, or off-policy with a training policy that is a mixture of $P_F$ with a uniform policy: $\tau = (s_0 \to s_1 \to \ldots \to s_n)$ is sampled with $s_{i+1} \sim (1 - \epsilon)P_F(s_{i+1}|s_i;\theta) + \epsilon\frac{1}{\#\{t:(s \to t) \in \mathcal{A}\}}$. Here $\epsilon$ is the random exploration weight.

## 2.3 SUBTRAJECTORY BALANCE: LEARNING FROM PARTIAL EPISODES

Recall the GFlowNet parametrization used in the DB objective above, with a state flow estimator $F(-|-;\theta)$ and a pair of policies $P_F(-|-;\theta), P_B(-|-;\theta)$. It is shown in §A.2 of Malkin et al. (2022) that the detailed balance conditions (6) are satisfied for all actions if and only if the following *subtrajectory balance* constraint holds for all (not necessarily complete) trajectories $\tau = (s_m \to \ldots \to s_n)$:

$$F(s_m;\theta)\prod_{i=m}^{n-1} P_F(s_{i+1}|s_i;\theta) = F(s_n;\theta)\prod_{i=m}^{n-1} P_B(s_i|s_{i+1};\theta), \tag{8}$$

where we again enforce that $F(x;\theta) = R(x)$ if $x$ is terminal. Observe that the DB condition (6) is a special case of (8) when the trajectory consists of one action, and the TB condition (7) is precisely the case when $\tau$ is complete, with the identification $Z_\theta = F(s_0;\theta)$.

The above constraint yields the *subtrajectory balance objective*

$$\mathcal{L}_{\text{SubTB}}(\tau) = \left(\log \frac{F(s_m;\theta)\prod_{i=m}^{n-1} P_F(s_{i+1}|s_i;\theta)}{F(s_n;\theta)\prod_{i=m}^{n-1} P_B(s_i|s_{i+1};\theta)}\right)^2. \tag{9}$$

If this objective is made equal to 0 for all partial trajectories $\tau$, where $R(s_n)$ is substituted for $F(s_n;\theta)$ if $s_n$ is terminal, then the policy $P_F$ satisfies the desired condition (2). (Proof: When $\mathcal{L}_{\text{SubTB}}(\tau) = 0$, (8) is satisfied, implying satisfaction of both (7) and (6). Either of these conditions is a sufficient condition for (2), as shown by Bengio et al. (2021b); Malkin et al. (2022).)

**Extracting subtrajectories for training.** Suppose that an episode (complete trajectory) $\tau = (s_0 \to s_1 \to \ldots \to s_n)$ is sampled for training. There are $\binom{n+1}{2} = O(n^2)$ nontrivial subtrajectories:

$$\tau_{i:j} := (s_i \to s_{i+1} \to \ldots \to s_j), \quad 0 \leq i < j \leq n. \tag{10}$$

Having sampled a complete trajectory $\tau$ for training, we make gradient steps on a convex combination of the subtrajectory balance losses $\mathcal{L}_{\text{SubTB}}(\tau_{i:j})$: $\theta \leftarrow \theta - \nabla_\theta \mathcal{L}$, where

$$\mathcal{L} = \frac{\sum_{0 \leq i < j \leq n} \lambda^{j-i} \mathcal{L}_{\text{SubTB}}(\tau_{i:j})}{\sum_{0 \leq i < j \leq n} \lambda^{j-i}}. \tag{11}$$

Here $\lambda > 0$ is a hyperparameter controlling the weights assigned to subtrajectories of different lengths, and when $\lambda$ is set to 1, it leads to a uniform weighting scheme. Notice that the $\lambda \to 0^+$ limit leads precisely to the average detailed balance loss $\mathcal{L}_{\text{DB}}(s_i \to s_{i+1})$ over all transitions in $\tau$, while the $\lambda \to +\infty$ limit gives the trajectory balance objective $\mathcal{L}_{\text{TB}}(\tau)$.[1]

Other schemes for weighting subtrajectories are possible and should be explored in future work.

**Computational considerations.** It may appear that the optimization of (11) induces a computation cost that is quadratic in the trajectory length. However, a closer inspection of the gradient of (11) with respect to the state flows $\log F(s_i;\theta)$ and the forward and backward policy logits shows that gradient computation requires only one forward and one backward pass through the neural networks giving $\log F(s;\theta)$, $\log P_F(-|s_i;\theta)$, and $\log P_B(-|s_i;\theta)$. The quadratic computation cost is incurred only in performing linear operations on these log-flows and policy logits, not in the evaluation of the deep networks. Thus the SubTB loss has little computation overhead over DB or TB.

---

[1]When a *batch* of trajectories is used for training, the convex combination weights may either be normalized over all subtrajectories of all trajectories in the batch, or normalized independently over the subtrajectories of each trajectory. For consistency, we choose the first option for the experiments in this paper.

**Hypothesized benefits.** We hypothesize that SubTB($\lambda$) brings two benefits to GFlowNet training:

VARIANCE REDUCTION. The TB loss terms $\mathcal{L}_{\text{TB}}(\tau)$ for trajectories $\tau$ that take a given sequence of actions until a state $s$, then diverge, share the terms $\log Z$ and the policy logits for all transitions preceding $s$ inside the square. However, the 'tail' of the TB loss, involving the forward and backward policy logits for transitions that appear after $s$ in $\tau$, can be seen as a *stochastic* least-squares regression target. That is, if $s = s_m$ in a trajectory $\tau = (s_0 \rightarrow s_1 \rightarrow \ldots \rightarrow s_n)$, then

$$\log\left(Z \cdot \prod_{i=0}^{m-1} \frac{P_F(s_{i+1}|s_i)}{P_B(s_i|s_{i+1})}\right) \tag{12}$$

is regressed to

$$\log\left(R(s_n) \cdot \prod_{i=m}^{n-1} \frac{P_B(s_i|s_{i+1})}{P_F(s_{i+1}|s_i)}\right). \tag{13}$$

Similarly, for trajectories that share the transitions *following* $s$ but may differ in their initial actions, (12) is a stochastic regression target for (13).

The subtrajectory balance loss terms $\mathcal{L}_{\text{SubTB}}(\tau_{m:j})$ for partial trajectories beginning at $s$ regress the log-state flow $\log F(s)$ to (parts of) expressions like (13), while loss terms $\mathcal{L}_{\text{SubTB}}(\tau_{i:m})$ regress (parts of) expressions like (12) to the log-state flow $\log F(s)$. The learned $\log F(s)$ is thus a learned estimate of a stochastic piece of the TB loss for trajectories that contain $s$. Replacing a stochastic term in the TB loss by a learned estimate of its expectation is guaranteed to introduce bias into the gradient (with respect to the gradient of the TB loss), but is expected to reduce variance. This is akin to the variance-reducing effect of actor-critic methods in RL.

This hypothesis is studied empirically in our experiments and in particular §4.1.1, where we provide evidence that SubTB($\lambda$) is a practically useful interpolation between TB (high variance) and DB (low variance, high bias relative to the true TB gradient) losses.

FASTER LEARNING DUE TO GENERALIZATION OF STATE FLOWS. Another benefit of subtrajectory balance for convergence speed may come from the ability of estimated state flow functions $\log F(s; \theta)$ to be modeled with high precision and generalize between states $s$ faster than the often high-dimensional policy logits $\log P_F(-|s; \theta), \log P_B(-|s; \theta)$. Such generalization is important in problems where the state graph becomes 'wide' far from the initial state, making the learning signal sparse at states that are near termination. Indeed, in all of our experiment domains except the hypergrids in §4.1 – and for the largest hypergrids – the number of terminal states is many orders of magnitude larger than the total number of states seen in training.

## 3 RELATED WORK

**Eligibility traces.** SubTB($\lambda$) draws inspiration from the TD($\lambda$) algorithm in RL (Sutton, 1988; Sutton & Barto, 2018), which forms an estimate of the expected return via a convex combination of $n$-step returns, each weighed by $(1 - \lambda)\lambda^{n-1}$. The parameter $\lambda \in [0, 1]$ enables a bias-variance tradeoff (Kearns & Singh, 2000). Intuitively, larger $\lambda$ leads to lower bias and higher variance, since the estimate of the expected return approaches the single-point Monte Carlo estimate as $\lambda \rightarrow 1$. We take inspiration from this idea to mix together different (possibly all) subtrajectories, akin to how $n$-step returns are mixed together. We hypothesize that the right mixing may reduce variance, compared to TB, with the additional benefits of inducing consistency between the flows of intermediate states, and thus of helping propagate credit faster and enable faster convergence. In addition, GFlowNet training objectives are reminiscent of *residual gradient* RL methods (Baird, 1995; Zhang et al., 2020) since the "endpoint" (e.g. $F(s_n)$ in (9)) is also considered in the gradient.

**MaxEnt RL.** RL has a rich literature on energy-based, or maximum entropy, methods (Ziebart, 2010; Mnih et al., 2016; Haarnoja et al., 2017; Nachum et al., 2017; Schulman et al., 2017; Haarnoja et al., 2018), which are close or equivalent to the GFlowNet framework in certain settings (in particular when the MDP has a tree structure (Bengio et al., 2021a)). Also related are methods that maximize entropy not on the policy, but rather on the state visitation distribution (Hazan et al., 2019; Islam et al., 2019; Zhang et al., 2021) or some proxy of it (Eysenbach et al., 2018), which achieve a similar objective to GFlowNet models by flattening the state visitation distribution. If the state graph of the environment is a directed tree, the loss $\mathcal{L}_{\text{SubTB}}$ on individual subtrajectories is equivalent to that of path consistency learning (Nachum et al., 2017). However, attempts to use path consistency learning in settings without intermediate rewards have only computed the loss on subtrajectories that have length 1 or include a terminal state (Guo et al., 2021).

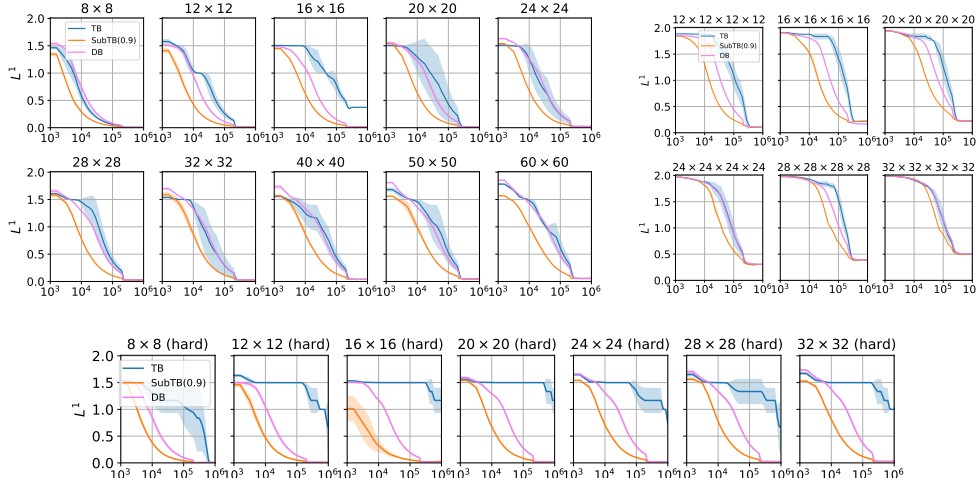

Figure 3: $L^1$ distance between empirical and target distributions over the course of training on the hypergrid environment. SubTB($\lambda = 0.9$) consistently gives faster convergence than TB, the strongest objective from past work, on all grid sizes. The difference is especially visible for the harder variant of the reward function (last row). The $x$-axis is the cumulative number of training trajectories (episodes).

## 4 EXPERIMENTS

### 4.1 HYPERGRID: ROBUSTNESS TO SPARSE REWARDS

We study the synthetic hypergrid environment introduced in Bengio et al. (2021a). The set of interior states is a $d$-dimensional hypergrid of size $H \times H \times \cdots \times H$ with a multimodal reward function concentrated near each of the $2^d$ corners of the hypergrid (see Bengio et al. (2021a); Malkin et al. (2022) and Fig. 1). The initial state is $(0, 0, \ldots, 0)$, and each action is a step that increments one of the $d$ coordinates by 1 without leaving the grid. A special termination action is also allowed from each state. This environment is designed to challenge a learning agent to infer and discover new modes from those that have been already been visited.

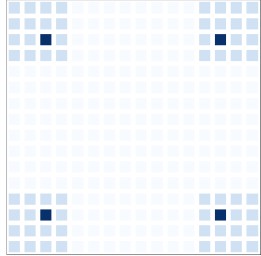

Figure 1: $16 \times 16$ hypergrid reward function.

We study various sizes of 2-dimensional and 4-dimensional hypergrids, using the hardest variant of the reward function from past work (the minimal reward, away from the corners of the grid, is set to $10^{-3}$). We train GFlowNets to sample from the target reward functions and plot the evolution of the $L^1$ distance between the target distribution and the empirical distribution of the last $2 \cdot 10^5$ states seen in training.[2] In all cases, we tune the learning rates for the TB and SubTB($\lambda = 0.9$) objectives. (See §A for details.)

The results (mean and standard deviation over three random runs) are shown in the first two rows of Fig. 3. Models trained with SubTB($\lambda$) converge faster, and with less variance between random seeds, to the true distribution than with TB for all hypergrid sizes.

We also study an even sparser variant of the environment, in which the background reward is set to $10^{-4}$. In this case, SubTB($\lambda$) continues to perform strongly (last row of Fig. 3), while models trained with TB do not even discover all modes of the target distribution for grids larger than $8 \times 8$ (Fig. 2).

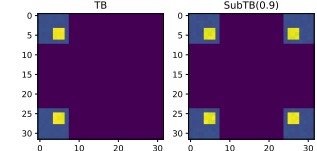

Figure 2: Distribution of $2 \times 10^5$ samples from GFlowNets trained on the harder variant of the $32 \times 32$ grid with TB and SubTB($\lambda$) objectives.

Additional results are given in §A.1. In particular, SubTB($\lambda$) continues to perform strongly when *only* subtrajectories of less than a certain length are used for training, which can be beneficial in realistic settings where only partial episodes are given. We also show the effect of $\lambda$ on the convergence rate (Fig. A.2) and of more exploratory training policies (Fig. A.3).

---

[2]Such an evaluation is possible in this synthetic environment because the exact target distribution function can be tractably computed. Note that the metric shown in Fig. 3 differs from what is called '$L^1$ distance' in past work, as we do not divide by the total number of states.

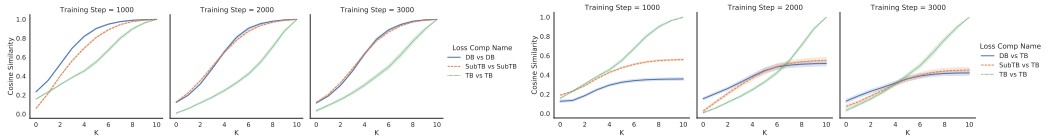

Figure 4: Mean cosine similarity between small-batch ($2^k$) and large-batch (1024) gradients at selected training iterations. **Left:** Small-batch vs. large-batch gradients of DB, SubTB($\lambda$), and TB objectives. **Right:** Small-batch DB, SubTB($\lambda$), and TB gradients vs. large-batch TB gradient.

### 4.1.1 A CLOSER LOOK AT GRADIENT VARIANCE

We take a closer look at gradient bias and variance to understand the benefits of training GFlowNets with SubTB($\lambda$). The methodology of these experiments is inspired by Ilyas et al. (2020).

We train GFlowNets on the $8 \times 8$ grid environment using SubTB($\lambda = 0.8$) and monitor various gradient metrics during training. To remove the effect of parameter sharing between policies at different states and to isolate the effect of the objective, we use a tabular representation of the GFlowNet, i.e., all flows and policy logits are optimized as independent parameters.

**Gradient variance.** To measure gradient variance, we use the following procedure for each training objective (DB, TB, or SubTB($\lambda$)). A large batch of $2^{10} = 1024$ trajectories is sampled, and the gradient $g_j^{(0)}$ of the objective with respect to the policy logits at all states is computed for each trajectory $\tau_j$ in the batch. Then, for each $k \in \{0, 1, \ldots, 9\}$, the gradients $g_i^{(0)}$ are combined into $2^{10-k}$ sub-batches, each of size $2^k$. The sub-batch gradient $g_i^{(k)}$ for the $i$-th sub-batch is set to the average of trajectory gradients $g_j^{(0)}$ contained within the sub-batch and computed for $i \in \{1, 2, \ldots, 2^{10-k}\}$. We then report the average cosine similarity between the sub-batch and full-batch gradients:

$$\frac{1}{2^{10-k}} \sum_{i=1}^{2^{10-k}} \frac{g_i^{(k)} \cdot g_1^{(10)}}{\left\| g_i^{(k)} \right\| \left\| g_1^{(10)} \right\|}.$$

If this quantity is positive, then gradient steps of infinitesimally small norm along the stochastic sub-batch gradient decrease the full-batch objective in expectation. Fig. 4 (left) shows the dependence of this metric on $k$ at various iterations. A steeper curve, such as those of DB and SubTB($\lambda$), indicates lower gradient variance.

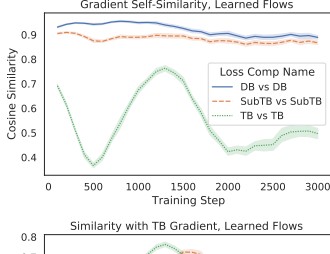

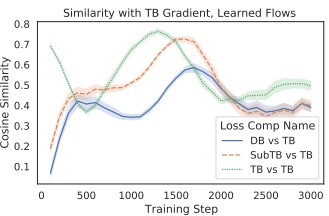

Figure 5: Mean cosine similarity between small-batch (64) and large-batch (1024) gradients on the $8 \times 8$ grid environment. **Above:** Self-similarity of the DB, SubTB($\lambda$), and TB gradients, showing DB < SubTB($\lambda$) < TB in gradient variance. **Below:** Similarity of small-batch DB, SubTB($\lambda$), and TB gradients to the large-batch TB gradient, showing that the small-batch SubTB($\lambda$) gradient is a good estimator of large-batch TB.

Fig. 5 (top) shows the metric at $k = 6$ (corresponding to the batch size of 64 used for training) over the course of training. We see that the DB gradient has the highest self-consistency at all iterations, TB has the lowest, and SubTB($\lambda = 0.8$) is in between.

**Gradient bias.** We next compare the small-batch stochastic gradients with large-batch stochastic gradients, using *different* objectives for the small and full batches. Specifically, we compare the small-batch DB, SubTB($\lambda$), and TB gradients with the full-batch TB gradient. (The full-batch TB gradient can be seen as a 'canonical' gradient against which bias can be measured, as its expectation equals the gradient of the KL divergence between the distribution over trajectories defined by $P_F$ and that defined by the reward $R$ and $P_B$; see §A.3 of Malkin et al. (2022).)

Fig. 5 (bottom) shows the cosine similarity at the batch size used for training. Notably, at intermediate iterations, the similarity of SubTB($\lambda$) with TB is higher than that of TB with TB: despite its bias, **the small-batch SubTB($\lambda$) gradient estimates the full-batch TB gradient better than the small-batch TB gradient does**. Fig. 4 (right) shows the dependence of the similarity on $k$ at selected iterations and suggests that this effect may be even larger for smaller batch sizes. Moreover, at $k = 10$, the similarity of SubTB($\lambda$) vs. TB always lies between DB vs. TB and TB vs. TB, indicating that SubTB($\lambda$) interpolates between TB's unbiased and DB's biased estimates of the TB gradient.

**The effect of learned state flows.** For additional experiments, see §A.2.

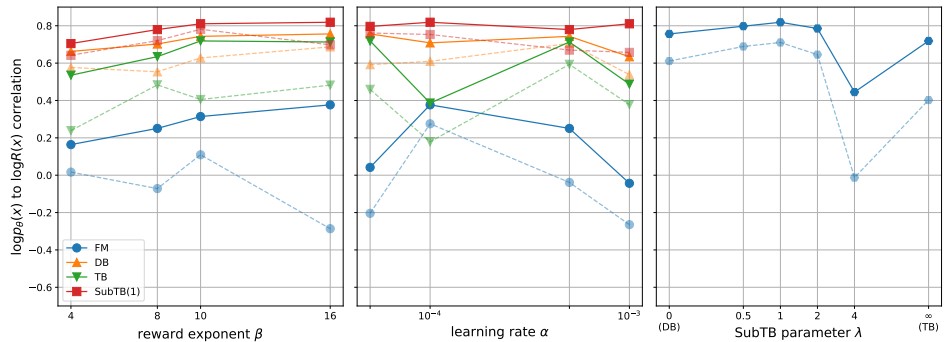

Figure 6: Correlation between marginal sampling log-likelihood and log-reward on the molecule task. For each hyperparameter setting on the $x$-axis, we plot the best result over choices of the other hyperparameter(s) – $\alpha$ in the left plot, $\beta$ in the centre plot, and both $\alpha$ and $\beta$ in the right plot – with a solid line. The mean result over values of other hyperparameter(s) is plotted with a dashed line.

## 4.2 SMALL MOLECULE SYNTHESIS

We use SubTB($\lambda$) to train models on the molecule generation task of Bengio et al. (2021a). The task is to generate binders of the sEH (soluble epoxide hydrolase) protein, based on a docking prediction (Trott & Olson, 2010). To be precise, molecules are generated by sequentially joining 'blocks' from a fixed library to the partial molecular graph (Jin et al., 2020; Kumar et al., 2012), resulting in a state space of estimated size $10^{12}$. The reward function $R$ is given by a pretrained proxy model made available by Bengio et al. (2021a). To adjust the greediness of the agent, an inverse temperature hyperparameter $\beta$ is used, i.e., the reward used for training is $R(x) = \widetilde{R}(x)^{\beta}$, where $\widetilde{R}(x)$ is the proxy's prediction.

We train models with the DB, TB, and SubTB($\lambda$) objectives, with four values each of $\lambda$, $\beta$, and learning rate, averaging the results over 3 random runs for each setting. We measure how well the trained models match the target distribution by the correlation of $\log R(x)$ and $\log p_{\theta}(x)$, the log-probability assigned to $x$ by the GFlowNet, computed on a held-out set of terminal states $x$.[3]

The results are shown in Fig. 6. SubTB($\lambda$), in particular with $\lambda = 1$, performs better than both DB and TB when the optimal hyperparameters $\alpha, \beta$ are used (solid lines) and is far more robust to the choice of hyperparameters (dashed lines). Additional details can be found in §B.

## 4.3 SEQUENCE GENERATION

We consider three sequence generation tasks in which sequences are generated left to right, with each action appending one symbol from a vocabulary to a partial sequence: a synthetic task with varying sequence lengths and vocabulary sizes (§4.3.1), a practical biological sequence design task (§4.3.2), and a new protein design task with longer sequences (4.3.3). For all three tasks, we consider the baselines Soft Actor-Critic (Haarnoja et al., 2018; Christodoulou, 2019), A2C with Entropy regularization (Williams & Peng, 1991; Mnih et al., 2016) and MARS-like MCMC (Xie et al., 2021) and compare them with three GFlowNet training objectives: TB, FM, and SubTB($\lambda$).

In §F, we also study a *non-autoregressive* sequence generation problem (inverse protein folding).

### 4.3.1 BIT SEQUENCES

We consider the synthetic sequence generation setting from Malkin et al. (2022), where the goal is to generate sequences of bits of fixed length $n = 120$. The reward is specified by a set of modes $M \subset \mathcal{X} = \{0, 1\}^{n}$ that is unknown to the learning agent. The reward of a generated sequence $x$ is defined in terms of Hamming distance $d$ from the modes: $R(x) = \exp(-\min_{y \in M} d(x, y))$.

The vocabulary size can be varied: for any integer $k$ dividing 120, we take a vocabulary consisting of words of length $k$ (so that the vocabulary size is $2^{k}$ and the full sequence is generated in $\frac{n}{k}$ actions). By varying the value of $k$ and keeping $n$ and $M$ constant, we study the behavior of learning agents with varying action space sizes and trajectory lengths without changing the underlying modeling problem. Most experiment settings are taken from Malkin et al. (2022); see §C.

---

[3]Comparing the exact sampling and target distributions, like in §4.1, is not possible here, since we cannot enumerate all terminal states. However, the marginal likelihood that a trained GFlowNet generates a given $x$ is tractable to compute by dynamic programming. For a model that samples perfectly from the target distribution, $\log(R(x))$ and $\log p_{\theta}(x)$ would differ by a constant $\log Z$ independent of $x$ and thus be perfectly correlated.

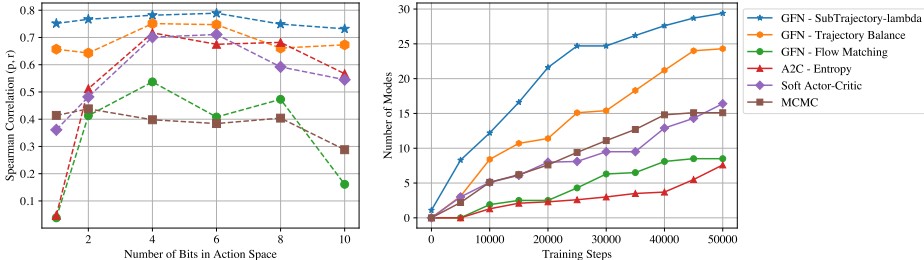

Figure 7: **Left:** For the number of bits $k \in \{1, 2, 4, 6, 8, 10\}$ in each vocabulary token, we plot the Spearman correlation between the sampling probability and reward on a test set for each method. Training with SubTB($\lambda$) leads to policies that have the highest correlation with the reward across all lengths and vocabulary sizes. **Right:** For $k = 1$, the number of modes discovered by each method over the course of training is plotted. SubTB($\lambda$) discovers more modes faster.

Models are evaluated by computing the Spearman correlation, on a test set of sequences $x$, between the probability of generating $x$ and the reward $R(x)$. We also track the number of modes discovered during the training process for all the methods, see Fig. 7. We find that models trained with the SubTB($\lambda$) objective have a higher Spearman correlation at the end of training and discover modes faster compared to the other GFlowNet objectives and non-GFlowNet baselines.

#### 4.3.2 Antimicrobial peptide generation

Next, we consider the task of generating peptides with antimicrobial properties (AMPs). These sequences have maximum length 60 and use a vocabulary of 20 amino acids (and an end-of-sequence token), resulting in a state space of size $21^{60}$. The reward function is a pretrained proxy neural network that estimates the antimicrobial activity. (See Jain et al. (2022) for details on this task.)

Table 2: Results on the AMP generation task (mean and standard error over 3 runs).

| Algorithm | Top-100 Reward | Top-100 Diversity |
|---|---|---|
| GFN-$\mathcal{L}_{\text{SubTB}(\lambda)}$ | 0.96 ± 0.02 | 42.23 ± 3.4 |
| GFN-$\mathcal{L}_{\text{TB}}$ | 0.90 ± 0.03 | 31.42 ± 2.9 |
| GFN-$\mathcal{L}_{\text{FM}}/\mathcal{L}_{\text{DB}}$ | 0.78 ± 0.05 | 12.61 ± 1.32 |
| SAC | 0.80 ± 0.01 | 8.36 ± 1.44 |
| AAC-ER | 0.79 ± 0.02 | 7.32 ± 0.76 |
| MCMC | 0.75 ± 0.02 | 12.56 ± 1.45 |

We train GFlowNets with the SubTB($\lambda$), TB, and FM losses and compare them with baselines. To evaluate the trained models, we sample 2048 sequences from the policy, then compute the mean reward and mean pairwise edit distance of the top-100 reward sequences. The metrics and model architecture are taken from Malkin et al. (2022); see §D. The results are presented in Table 2. SubTB($\lambda$) provides significant improvements over all the baselines (including TB, FM, and DB GFlowNets) in both reward and diversity.

#### 4.3.3 Fluorescent protein generation

We consider the task of generating protein sequences with fluorescence properties (Trabucco et al., 2022) to evaluate SubTB($\lambda$) in settings with longer trajectories. In this task, sequences have a fixed length of 237, and the size of the state space is $20^{237}$. The proxy reward function $R(x)$ is trained on a dataset of proteins with their fluorescence scores from Sark-isyan et al. (2016). The metrics and models are the same as in §4.3.2; see §E for details.

Table 3: Results on the GFP generation task (mean and standard error over 3 runs).

| Algorithm | Top-100 Reward | Top-100 Diversity |
|---|---|---|
| GFN-$\mathcal{L}_{\text{SubTB}(\lambda)}$ | 1.18 ± 0.10 | 204.44 ± 0.45 |
| GFN-$\mathcal{L}_{\text{TB}}$ | 0.76 ± 0.19 | 204.31 ± 0.44 |
| GFN-$\mathcal{L}_{\text{FM}}/\mathcal{L}_{\text{DB}}$ | 0.30 ± 0.08 | 190.21 ± 6.78 |
| SAC | 0.23 ± 0.03 | 120.32 ± 15.57 |
| AAC-ER | 0.22 ± 0.02 | 113.65 ± 21.31 |
| MCMC | 0.28 ± 0.01 | 169.17 ± 12.44 |

The GFlowNet objectives outperform all other methods in both metrics, finding more diverse and higher-reward sequences (Table 3). SubTB($\lambda$) significantly outperforms TB, while achieving a similar diversity. We note that the advantage of SubTB($\lambda$) is greater than that in the AMP task (Table 2) and speculate that the benefits of SubTB($\lambda$) become more prominent for longer action sequences.

## 5 Discussion and conclusion

We have given evidence of a bias-variance tradeoff in GFlowNet training algorithms. The high-variance stochastic regression objective of TB and the low-variance local consistency objective of DB lie at opposite ends of this range. We showed that SubTB($\lambda$) can harness the variance-reducing effects of local objectives while retaining the fast credit assignment properties of trajectory-level objectives. We see *learnable* strategies for selecting and weighting (sub)trajectories for training – e.g., a dynamic choice of $\lambda$ and an active-learning approach to sampling trajectories – as the most interesting questions for future work. The ability of subtrajectory objectives to learn from incomplete episodes also makes their application in RL environments an appealing research direction.

## REPRODUCIBILITY STATEMENT

We provide extensive experiment details, such as learning rates, batch sizes, number of training steps, choices of $\lambda$, description of attempted hyperparameters, and additional clarifying experiments in the Appendices. Code for experiments on the hypergrid domain (§4.1) and on the molecule domain (§4.2) is also provided with the submission.

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

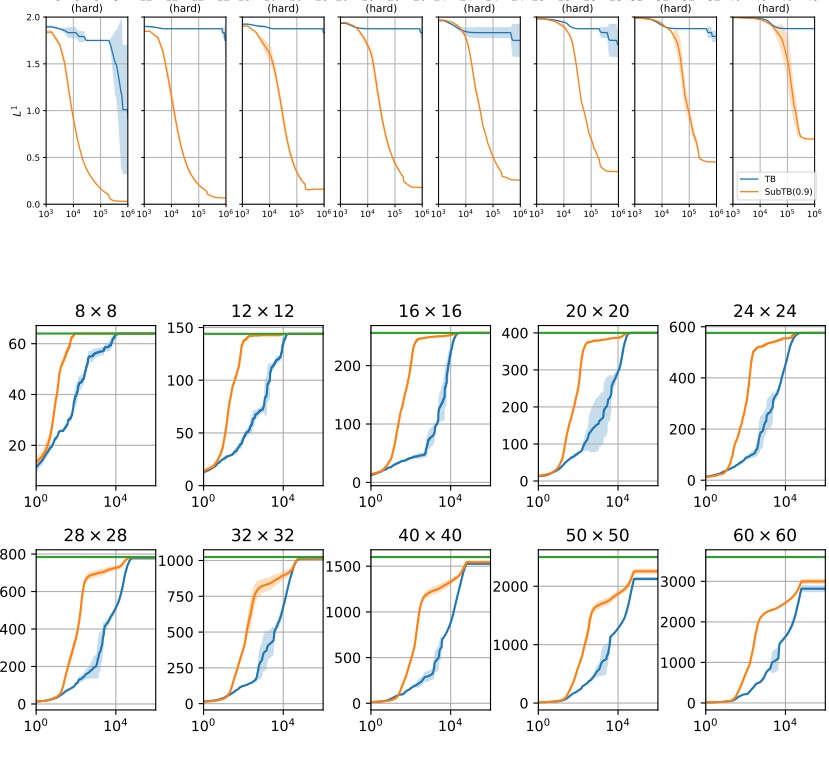

Figure A.1: Additional results for hypergrid experiments. **Above:** The evolution of the $L^1$ between empirical sampling and target distributions on the harder variants of 4-dimensional grids, in the same format as Fig. 3. **Below:** The number of cumulative distinct terminal states visited as a function of training time on the standard 2-dimensional grid. Models trained with SubTB($\lambda$) discover more states faster.

## A EXPERIMENT DETAILS: HYPERGRID

The environment is identical to that in Malkin et al. (2022), with reward function parameters $(R_0, R_1, R_2) = (10^{-3}, 0.5, 2)$ for the standard variant of the grid and $(10^{-4}, 1.0, 3.0)$ for the harder variant. The models giving logits of $P_F(-|s)$ and $P_B(-|s)$, as well as $\log F(s)$, are MLPs of the same architecture as in Bengio et al. (2021a), taking a one-hot representation of the coordinates of $s$ as input and sharing all layers except the last. The initial state flow $\log Z = \log F(s_0)$ is an independent parameter whose learning rate is set to $10\times$ the learning rate of other parameters.

All models are trained with the Adam optimizer and a batch size of 16 for a total of $10^6$ trajectories (62500 batches). The optimal learning rate for each experiment is chosen from $\{0.0005, 0.00075, 0.001, 0.003, 0.005, 0.0075, 0.01\}$, and $\lambda = 0.9$ is chosen as the optimal value from the set $\{0.8, 0.9, 0.99\}$.

Gradient bias and variance experiments are conducted in the harder variant of the $8 \times 8$ grid. The tabular GFlowNet is trained using Adam with a learning rate 0.007 and the SubTB($\lambda = 0.8$) objective.

### A.1 ADDITIONAL EXPERIMENTS

Fig. A.1 shows additional results on more difficult grid environments.

We perform another experiment in which only short (up to length 4) subtrajectories are used for training with the SubTB($\lambda$) objective (i.e., the sum in (11) is truncated to exclude pairs $(i, j)$ with $j - i > 4$). The results, shown in Fig. A.4, show that SubTB($\lambda$) continues to perform strongly in this restricted setting.

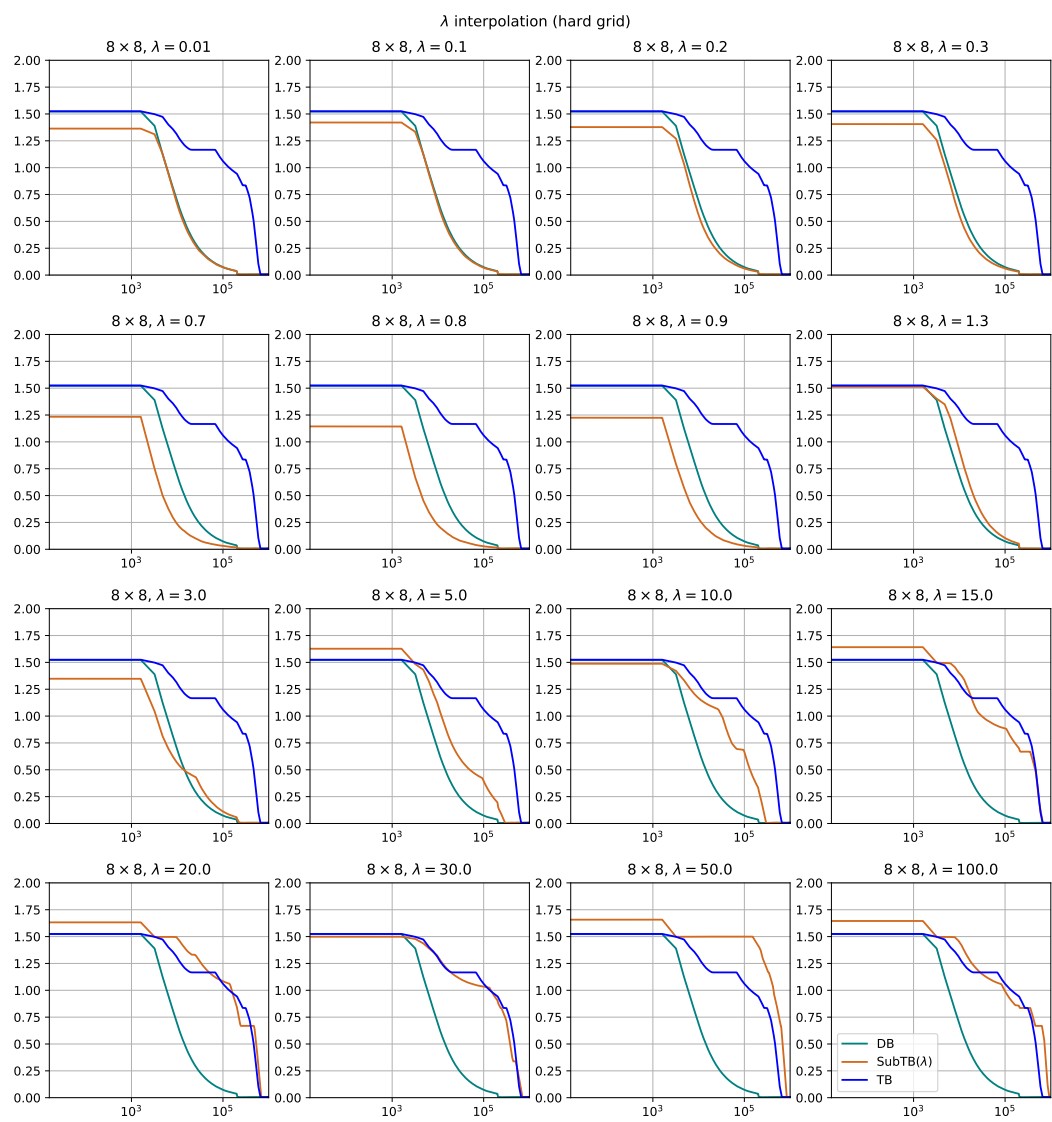

Figure A.2: Empirical $L^1$ curves on the $8 \times 8$ grid for varying values of $\lambda$.

Fig. A.2 shows the effect of the SubTB parameter $\lambda$ on the training curves, showing a gradual interpolation between DB and TB and fastest convergence at values slightly less than 1.

Fig. A.3 contains visualizations of the exploration behavior of different training algorithms. It shows that TB can perform better with off-policy training and can benefit from a higher temperature of the policy logits, but still does not learn as fast as SubTB($\lambda$), nor does it find all the modes in the maximum number of training iterations.

## A.2 MORE ON BIAS AND VARIANCE: THE EFFECT OF LEARNED STATE FLOWS

To better understand the variance-reducing properties of SubTB($\lambda$), we perform the gradient bias experiments with a modified computation of gradients that removes the factor of learning the state flows.

Recall from §2.1 that a forward policy $P_F$ uniquely determines a distribution over trajectories. If the initial state flow $Z$ and forward policy $P_F$ are fixed, there is a unique state flow function $F^F$ and backward policy $P_B$ that satisfy the detailed balance conditions (6). This 'true forward' flow function, written $F^F(s) = Z \sum_{\tau:s\in\tau} P_F(\tau)$, is determined by an initial state flow fixed to the true

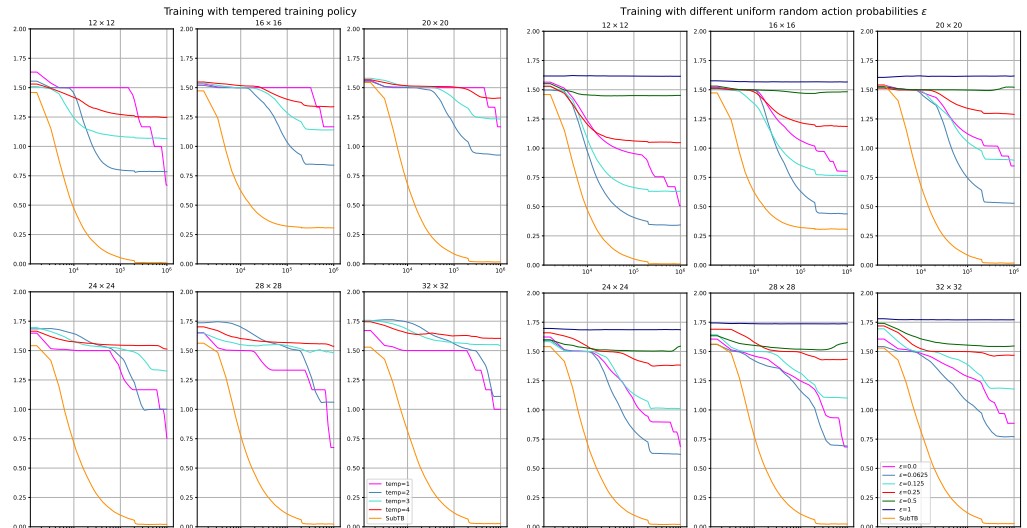

Figure A.3: Training GFlowNets on the harder variants of 2-dimensional grids using a tempered training policy (**left**), and a training policy that takes a uniformly random action with probability $\epsilon$ at each sampling step (**right**).

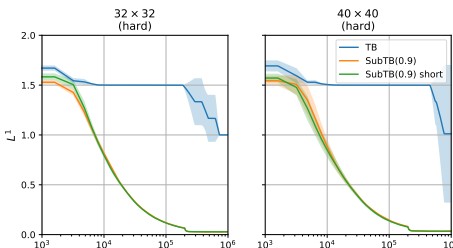

Figure A.4: Training GFlowNets using only short subtrajectories in different grid environments using the SubTB($\lambda = 0.9$) objective.

partition function $Z = \sum_{x \in \mathcal{X}} R(x)$ and the learned forward policy $P_F$. Similarly, the 'true backward' flow function, written $F^B(s) = \sum_{\tau : s \in \tau} P_B(\tau) R(x_\tau)$ where $x_\tau$ is the terminal state of $\tau$, is uniquely determined by the reward function $R$ and the learned backward policy $P_B$. In particular, $F^B(s_0) = \sum_{x \in \mathcal{X}} R(x)$.

We repeat the experiments on gradient bias, but by replacing the learned state flows $F$ in the losses by either the true forward or the true backward state flows ($F^F$ or $F^B$ respectively) computed exactly using the current values of the learned $P_F$ and $P_B$. (These modifications are not applied in training, but are used only to compute the gradient similarities. The small size of the environment makes computation of the true state flows tractable; this is not possible in general.)

The gradient similarity over the course of training is shown in Fig. A.5 (cf. Fig. 5 in the main text). The similar behavior of SubTB($\lambda$) with learned and true forward state flows suggests that the learned state flows remain close enough to their optimal values and that the variance-reducing benefits of SubTB($\lambda$) with true state flows are retained.

## B    EXPERIMENT DETAILS: MOLECULES

All experiments with SubTB($\lambda$) are based upon the published code of Malkin et al. (2022), which extends that of Bengio et al. (2021a). The proxy model giving the reward, the held-out set of molecules used to compute the correlation metric, and the GFlowNet model architecture – a graph neural network – are identical to those in Bengio et al. (2021a), and the off-policy exploration rate and early stopping likelihood are the same as those tuned for the training with the TB objective in

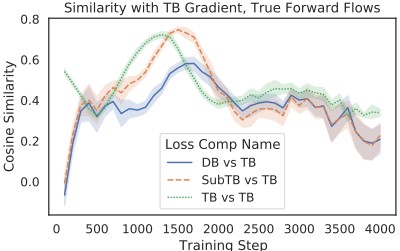
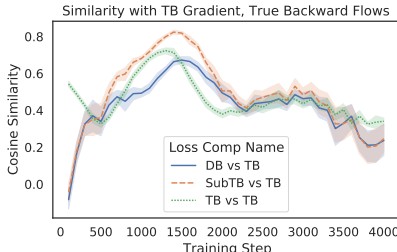

Figure A.5: Gradient similarity with state flows analytically computed in two ways (see §A.2). (Compare with Fig. 5.)

Malkin et al. (2022). All models are trained for a maximum of 50000 batches of 4 trajectories each. (Some training runs terminated early because of numerical overflows in the gradients, in which case we report the metric of the last stable model whose cumulative number of batches is a multiple of 5000.)

## C  EXPERIMENT DETAILS: BIT SEQUENCES

The modes $M$ as well as the test sequences are selected as described in  Malkin et al. (2022). The policy for all methods is parameterized by a Transformer (Vaswani et al., 2017) with 3 layers, dimension 64, and 8 attention heads. All methods are trained for 50,000 iterations with minibatch size of 16 using Adam optimizer. For GFlowNets with FM objective as well as the baselines, we use the exact same implementation and hyperparameters reported in Malkin et al. (2022). For TB and SubTB($\lambda$), we pick the best learning rate from $\{0.0075, 0.001, 0.001, 0.003, 0.005\}$ for forward logits, and for Z, use a learning rate of $10\times$ the learning rate for forward logits. For SubTB($\lambda$), we found the best $\lambda$ value of 1.9 from the values $\{0.8, 0.9, 1.1, 1.3, 1.5, 1.7, 1.9, 2.0\}$.

## D  EXPERIMENT DETAILS: ANTIMICROBIAL PEPTIDE GENERATION

Following Malkin et al. (2022) we use the following amino acids: [`A', `C', `D', `E', `F', `G', `H', `I', `K', `L', `M', `N', `P', `Q', `R', `S', `T', `V', `W', `Y']. We take 6438 known AMP sequences and 9522 non-AMP sequences from the DBAASP database Pirtskhalava et al. (2021). The classifier that serves as the proxy reward function is trained on this dataset, using 20% of the data as the validation set. The reward model is a Transformer, with 4 hidden layers, hidden dimension 64, and 8 attention heads. We train it with a minibatch of size 256, with learning rate $10^{-4}$, and with early stopping on the validation set. We use a Transformer with 3 hidden layers with hidden dimension 64 with 8 attention heads as the architecture of the policy for all methods. All methods are trained for 20,000 iterations, with a minibatch size of 16, using the reported hyperparameters for all the baselines from (Malkin et al., 2022). For TB and SubTB($\lambda$), we pick the best learning rates from $\{0.005, 0.007, 0.01, 0.03, 0.05, 0.07\}$ for forward logits and from $\{0.007, 0.01, 0.03, 0.05\}$ for log Z. For SubTB($\lambda$), the best performing $\lambda$ value of 1.9 chosen from $\{0.9, 0.99, 1.1, 1.2, 1.3, 1.4, 1.6, 1.7, 1.8, 1.9, 2.0\}$ is used.

## E  EXPERIMENT DETAILS: FLUORESCENT PROTEIN GENERATION

We consider a variant of the GFP task from Trabucco et al. (2022).  The vocabulary of amino acids is the same as §D: [`A', `C', `D', `E', `F', `G', `H', `I', `K', `L', `M', `N', `P', `Q', `R', `S', `T', `V', `W', `Y']. Following Trabucco et al. (2022), we consider the dataset of 56,086 proteins from Sarkisyan et al. (2016) processed based on Brookes et al. (2019). Each protein is accompanied by a score quantifying its fluorescence. As with the AMP data, we keep 20% of the data as a validation set used for early-stopping. The regressor trained with the dataset is a Transformer, with 4 hidden layers, hidden dimension 64, and 8 attention heads. We train it with a minibatch of size 256, with learning rate $10^{-4}$, with early stopping on the validation set. The architecture of the policy for all methods is a Transformer with 3 hidden layers with hidden dimension 64 with 8 attention heads. All methods are trained for 20,000 iterations, with a minibatch size of 16. We use the same implementation for all methods as the ones used in §D.

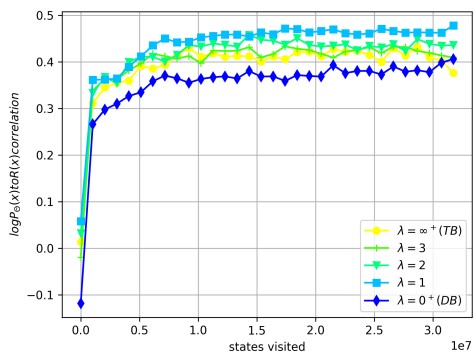

Figure F.1: The Spearman correlation between the sampling probability and reward on a test set is plotted over the course of training for each value of $\lambda$.

To define an exploratory training policy, we set the the random action probability to 0.01 selected from $\{0.0001, 0.0005, 0.001, 0.01\}$ and the reward exponent $\beta$ (having the same meaning as in §4.2) to 3 selected from $\{2, 3, 4\}$. For trajectory balance we use a learning rate of $5 \times 10^{-3}$ selected from $\{10^{-5}, 10^{-4}, 5 \times 10^{-4}, 10^{-3}, 5 \times 10^{-3}\}$ for the flow parameters and $1 \times 10^{-2}$ for $\log Z$. For SubTB($\lambda$), we choose the best $\lambda$ from $\{0.7, 0.8, 0.9, 0.99\}$, and found $\lambda = 0.99$ to perform the best. For TB and SubTB($\lambda$), we tune for the best learning rates from $\{0.0001, 0.0003, 0.0005, 0.00075, 0.001\}$ for the forward logits. For $\log Z$, we use a learning rate of $10\times$ the learning rate for the forward logits.

For FM we use a learning rate of $10^{-3}$ selected from $\{10^{-5}, 10^{-4}, 5 \times 10^{-4}, 10^{-3}, 5 \times 10^{-3}\}$ with leaf loss coefficient $\lambda_T = 30$. For A2C with entropy regularization we share parameters between the actor and critic networks, and use learning rate of $5 \times 10^{-3}$ selected from $\{10^{-5}, 10^{-4}, 5 \times 10^{-4}, 10^{-3}, 5 \times 10^{-3}\}$ with entropy regularization coefficient $5 \times 10^{-2}$ selected from $\{10^{-4}, 10^{-3}, 5 \times 10^{-3}, 10^{-2}, 5 \times 10^{-2}\}$. For SAC we use the formulation in Christodoulou (2019) with a learning rate of $10^{-3}$ selected from $\{10^{-5}, 10^{-4}, 5 \times 10^{-4}, 10^{-3}, 5 \times 10^{-3}\}$, a target network update frequency of 400 and initial random steps of 200. For the MARS baseline, we set the learning rate to $5 \times 10^{-4}$ selected from $\{10^{-5}, 10^{-4}, 5 \times 10^{-4}, 10^{-3}, 5 \times 10^{-3}\}$. We run the experiments on 3 seeds and report the mean and standard error over the three runs in Table 3.

## F INVERSE PROTEIN FOLDING: NON-AUTOREGRESSIVE SEQUENCE GENERATION

We consider the inverse protein folding problem suggested in Sinai et al. (2020). A target protein 3D backbone conformation is given, and the task is to sample amino acid sequences of a fixed length $L = 40$ from the Boltzmann distribution corresponding to their energy in the target conformation. The energy is provided by a physics model (Rohl et al., 2004; Chaudhury et al., 2010). The policy model is a 3-layer convolutional architecture that closely follows previous work (Sinai et al., 2020). Specifically, for the policy function, the convolution size was set to 7 with 32 hidden features and ReLU activation in each layer. The policy network has one additional convolutional layer of size 20 (number of amino acids), and without the activation function. The flow network has an additional two linear layers of sizes [1280,64], and [64, 1] with ReLU activation in between. We report mean result over three runs.

For this task, rather than generating sequences from left to right, we consider an action space in which actions modify one letter at a time at arbitrary positions. The first action uniformly randomly samples an amino acid sequence. On each subsequent action, the agent selects a position in the sequence and replaces the letter in this position with another letter in the vocabulary. Generation terminates after exactly $N = 40$ replacement steps. The forward policy is conditioned on the number of steps taken so far in the trajectory; the backward policy is fixed to be uniform over the $N \cdot L$ actions.

As a metric of how well the learned model matches the target distribution, we measure the correlation between $\log R(x)$ and the marginal sampling likelihood $\log p_\theta(x)$ on a held-out set of terminal

states. The results are presented in Fig. F.1. We observe that intermediate values of lambda lead to the best fit to the target distribution.

