# OpenReview forum: "Learning GFlowNets from partial episodes for improved convergence and stability"
_ICLR.cc/2023/Conference — Submitted to ICLR 2023_

### Official Review · Reviewer_aGGf · 2022-10-20

**Confidence:** 3
**Correctness:** 3
**Technical Novelty And Significance:** 2
**Empirical Novelty And Significance:** Not applicable
**Recommendation:** 5

**Clarity, Quality, Novelty And Reproducibility:**



The clarity of the authors work would be helped if figure 3 specified what the x axis is (number of steps, number of episodes...)

I don't find the experiments to be reproducible. Even if the authors just made a slight modification to Malkin et al. (2022), they don't discuss how they mixed P_F with a uniform policy for exploration, i.e. the constant they chose for \epsilon. Other key details, such as the general architecture of their estimator are also missing.

**Strength And Weaknesses:**

Strengths:

- well written, structured and easy to follow
- idea is plausible and understandable
- experiments improve on state of the art

Weaknesses (in no particular order)

Novelty+Relevance: the paper uses a slight modification on Trajectory balance objective, replacing whole trajectories with sub-trajectories. In the paper, the authors show how it provides better experimental results but don't argue how it advances theoretical understanding of GFlowNets. This issue is compounded since it's work on GFlowNets, which have such strong assumptions (must not contain cycles) that it's not applicable to most problems (chess, go, atari...)

Soundness: The authors claim the improvements they show in experimental results come from their novel SubTB method, but there seems to be another key difference between their work and Malkin et al. (2022). In Malkin, they claim they facilitate exploration by choosing a policy based on a "tempered (higher temperature) version" of P_F, while the authors use a "training policy that is a mixture of P_F with a uniform policy." It becomes unclear if gains are due to a different exploration policy or SubTB.

Another point that is unclear is the following: by using the criteria from eq. 11 with the O(n^2) different subsequences, the gradient's variance is reduced (as the authors claim), allowing for a larger learning rate (0.007 vs. 0.001 in Malkin et al. (2022)). The question is the following: if something is done to reduce the variance of TB, (for example using batches from a replay buffer) so that TB can be trained with a larger learning rate, will the author's method still be better? So is it the sampling of sub-sequences that makes the difference, or the variance reduction that comes from averaging the loss from lots of sequences.



**Summary Of The Paper:**

This paper discusses an improved training procedure to solve GFlowNet problems. The paper begins by outlining GFlowNets, a DAG with exactly one root node where "actions" are decisions about which edge to follow in the DAG. When the agent reaches a terminal node (a node with no edges leaving that node), the "episode" ends. The goal is to find a policy such that the probability of reaching a terminal state is proportional (up to a constant) to a reward function R.

GFlowNets have strong assumptions not in MDPs (absence of cycles) and a different objective (probability of reaching terminal states), the authors review previous work which shows one can solve the GFlowNets problem with a solution involving an edge flow function which satisfies certain constraints.

From there the authors outline previous methods used to solve the edge flow problem, such as Detailed balance and Trajectory balance all of which involve minimizing an objective, which once minimized is guaranteed to result in a policy matching the desired distribution.

The authors note that in  Malkin et al. (2022), it is shown that a flow solves the balance equation from the DB method if it holds for sub-sequences of nodes in trajectores. Using this observation, the authors present a new objective subtrajectory balance objective, resulting in their method.

They then hypothesize on why their new method brings a benefit.

The authors then repeat the experiments from Malkin et al. (2022), showing better results and an experiment on gradient variance and gradient bias.




**Summary Of The Review:**

I do not recommend this paper for publication, since it's only a small modification of previous work that doesn't advance the reader's understanding of science. Plus, there's an array of open questions about why the author's method delivers the observed performance gains.

---

> ### Author Response · Authors · 2022-11-09
> **Authors' response I: Answering questions**
>
> Thank you for your good questions and constructive feedback.
>
> > Novelty+Relevance: the paper uses a slight modification on Trajectory balance objective, replacing whole trajectories with sub-trajectories. In the paper, the authors show how it provides better experimental results but don't argue how it advances theoretical understanding of GFlowNets.
>
> The previous GFlowNet training objectives are either based on using a single transition (Detailed Balance) or using the full trajectory (Trajectory Balance) to update the policy parameters. This leaves out the flexibility of being able to learn from partial subtrajectories of any length. At the same time, fixing the length of these subtrajectories could immensely limit on what kind of subtrajectories could be used for learning. Thus, in this work, to remove this limitation, we proposed a novel objective which provides capability to not only efficiently learn from sub-trajectories of any length or all lengths, but also enables much faster convergence. (Note that although [Malkin et al., 2022] stated the SubTB constraint, they did not discuss the question of how to select subtrajectories for training or suggest to use a hyperparameter $\lambda$ to interpolate between DB and TB.)
>
> Please also see the answer to all reviewers on theoretical analysis.
>
> > The clarity of the authors work would be helped if figure 3 specified what the x axis is (number of steps, number of episodes...)
>
> The x-axis is the number of episodes. We have added this to the caption; we agree this will help the reader.
>
> > This issue is compounded since it's work on GFlowNets, which have such strong assumptions (must not contain cycles) that it's not applicable to most problems (chess, go, atari...)
>
> First, we want to mention that the assumption of acyclicity is something which is related to any GFlowNet learning objective and not specifically to the SubTB($\lambda$) objective introduced in this work.
>
> Second, this paper, and earlier GFlowNet papers, do not claim that they are a stand-in replacement for RL algorithms in general. GFlowNets should really be thought as inference machines or generative models rather than a general-purpose RL method. Standard RL could be used to generate objects sequentially as well, but with a different objective (maximizing the reward rather than sampling proportionally to the reward). Sampling is more appropriate when we are trying to do probabilistic inference. That being said, the paper "GFlowNet Foundations" [Bengio et al., 2021b] shows how one can incorporate sequences actions that can go back and forth, e.g. adding or removing pieces, increasing and decreasing coordinates, etc, if one adds an extra variable to the state, such as a time stamp, which always increases after each action, thus guaranteeing that the set of trajectories still forms a DAG. (It can also be shown that much of the GFlowNet theory continues to hold even if the state graph contains cycles, under mild assumptions such as nonvanishing of the policy, but this is beside the point of this paper.)

---

> ### Author Response · Authors · 2022-11-09
> **Authors' response II: Correcting a misunderstanding on reproducibility**
>
> > The authors claim the improvements they show in experimental results come from their novel SubTB method, but there seems to be another key difference between their work and Malkin et al. (2022). In Malkin, they claim they facilitate exploration by choosing a policy based on a "tempered (higher temperature) version" of P_F, while the authors use a "training policy that is a mixture of P_F with a uniform policy." It becomes unclear if gains are due to a different exploration policy or SubTB.
>
> In fact, there is no difference: in [Malkin et al., 2022], GFlowNets on the grid environment were trained on-policy, just as in our work, and GFlowNets trained on molecules used a mixture of the policy with uniform, just as in our work. (Our experiments are based upon the published code of [Malkin et al., 2022]; the comment about tempering the policy in the theory section of that paper, which is qualified by the word "usually", does not apply to their experiments.)
>
> As the GFlowNets trained with Trajectory Balance on the hypergrid environment fail to find all modes of the reward distribution, we also perform an ablation of exploration strategies applied to the policy trained via TB in Appendix A.1. Figure A.3 shows that even with the exploration strategies used in [Malkin et al. (2022)], training on-policy with SubTB outperforms training with TB.
>
> > Another point that is unclear is the following: by using the criteria from eq. 11 with the O(n^2) different subsequences, the gradient's variance is reduced (as the authors claim), allowing for a larger learning rate (0.007 vs. 0.001 in Malkin et al. (2022)). The question is the following: if something is done to reduce the variance of TB, (for example using batches from a replay buffer) so that TB can be trained with a larger learning rate, will the author's method still be better? So is it the sampling of sub-sequences that makes the difference, or the variance reduction that comes from averaging the loss from lots of sequences.
>
> In order to properly evaluate both TB and SubTB, we tried a large number of learning rates for all methods, and full details can be found in the respective section for each task domain in the Appendix. We found that larger learning rates do not always help TB. We do not use replay buffers for any of the settings, as off-policy learning methods is not the main point of the current work and we have not yet found improvements using them. Our ablation results on using other exploration schemes for Trajectory Balance emphasize that the benefits of the proposed method indeed come from the variance reduction properties of SubTB($\lambda$), which outperforms previous methods across all domains.
>
> > I don't find the experiments to be reproducible. Even if the authors just made a slight modification to Malkin et al. (2022), they don't discuss how they mixed P_F with a uniform policy for exploration, i.e. the constant they chose for \epsilon. Other key details, such as the general architecture of their estimator are also missing.
>
> As noted above, for the hypergrid and molecule tasks, there is no modification to the training policy or architectures from [Malkin et al., 2022]. For all other tasks, we ask you to refer to the specific section of the Appendix for each dataset, in which we provide the full architecture details along with the sets and ranges of the hyperparameters used.
>
> However, thanks to the reviewer's comment, we realized that we neglected to provide the architecture details for the inverse protein folding task. We have added these details to Appendix F.

---

> ### Author Response · Authors · 2022-11-17
> **Follow-up**
>
> We would like to thank the reviewer again for their useful insights and comments. We have addressed and responded to all of the points mentioned in the review, a quick summary of which is below:
>
> 1. **(a) how does the proposed work advance theoretical understanding of GFlowNets. (b) concern on strong assumptions related to GFlowNets:**
> We discuss the theoretical aspects of the proposed method under our reply titled "On Theoretical analysis of SubTB", in which we explain how and why we considered using an empirical formulation for the current method, SubTB($\lambda$) as well as to discuss the reason behind potential benefits through an exhaustive gradient analysis of all methods (Section 4.1.1). We also use the empirical experiments to support the motivation and formulation behind SubTB($\lambda$) to depict how the $\lambda$ parameter interpolates between the previous methods (Fig. A.2). The concern related to the assumptions of GFlowNets are not specific to the proposed method and independent of the current work. We hope our response was useful in providing some more context on the approach used in our paper. In addition, the proposed method is not just an extension of the previous work on Trajectory Balance, but rather a more general formulation of the both of the existing GFlowNet Training objectives of Detailed Balance (DB) and Trajectory Balance (TB).  Essentially, the proposed method of SubTB($\lambda$) provides a more general formulation through which DB and TB can be extracted through the hyperparameter $\lambda$ (see Section 2.3, “Extracting subtrajectories for training”), as well as supporting results on this smooth interpolation between DB and TB in Fig. A.2.
>
> 2. **Details on if gains are due to a different exploration policy or SubTB:**
> We discuss this in our previous response and also provide full details on the learning rates and other hyperameter settings and architecture used in the Appendix. We hope this helps.
>
> 3. **Details on plots, choice of exploration policy and architecture:**
> We have added the details to our plots, as mentioned in our previous response. While all other hyperparameters are fully specified in the Appendix, we have added the architecture missing details in one of our methods in our current submission. We hope this will help with the reader in understanding our choices better.
> In addition, we would again like to specify that the proposed method is not just an extension of Trajectory Balance, but an extension of the previous GFlowNet objectives of TB and DB as SubTB($\lambda$) provides a general formulation of these training objectives through the hyperparameter $\lambda$. Please refer to the specific pointers in this regard to this in the discussion above.
>
>
> 4. **Relation to the previous work and discussion on the observed performance gains:**
> We hope the details that we provided above on how the proposed method provides a general formulation of the previous GFlowNet training objectives are helpful in clarifying the benefits of SubTB($\lambda$) in which the $\lambda$ hyperparameter can be used to derive both DB and TB. This smooth transition provided by the proposed method, SubTB($\lambda$), also advances understanding of TB and DB, and is able to achieve the best characteristics of the previous objectives by improving convergence and reducing variance at the same time, and thus providing a much more robust and better training objective for GFlowNets. In addition, we show that using the proposed SubTB($\lambda$) objective, GFlowNets can be trained on harder and sparser reward configurations, as well as on partial sub-trajectories of any length, which was not possible before.
>
> We are enthusiastic to improve our work and welcome any more suggestions and feedback that the reviewer might have to help us in this direction. Thank you again for your time.

---

### Official Review · Reviewer_dWW3 · 2022-10-23

**Confidence:** 3
**Correctness:** 3
**Technical Novelty And Significance:** 3
**Empirical Novelty And Significance:** 3
**Recommendation:** 5

**Clarity, Quality, Novelty And Reproducibility:**

**Clarity**

The paper is clear and well-written. Some suggestions are provided above to make the paper more self-contained.

**Quality**

The technical content of the paper is correct and an extensive experimental evaluation is conducted in order to support the proposed idea.

**Novelty**

The authors propose a new objective unifying existing criteria used to train GFlowNets. The idea is inspired by unifying attempts in reinforcement learning between temporal difference and Monte Carlo estimators. An extensive experimental evaluation is conducted to fill the missing gap on the theoretical side.

**Reproducibility**

Code is made available in the submission. I haven’t run it to check its reproducibility though.

**Strength And Weaknesses:**

**Strenghts**
- The paper is well-written. However additional details can be included in order to make it more self-contained (see comments later)
- The idea is novel, original and the technical content is correct.
- Several tasks are used for experimental evaluation.
- Sequence generation tasks provide convincing evidence.
- Code is available. However, I haven’t run it to check if experiments are reproducible.

**Weaknesses**
- The sampling strategy doesn’t come with statistical guarantees on the quality of the obtained samples (like MCMC does). However, this is an intrinsic limitation of GFlowNets rather than of the proposed strategy.
- No supporting theory is developed to analyze the bias-variance tradeoff.
- Some experiments (hypergrid, bit sequences) miss baselines. Additionally, some experiments miss explanations of observed phenomena. Please refer to subsequent comments.

**Clarifying questions to improve the quality of the paper**

About training details:
- Can you provide an algorithmic table to describe the training of GFlowNets and the computation of the proposed objective in order to make the paper self-contained? Additionally, is the partial trajectory objective computed at the end of or during each episode in a TD-like fashion?

About hypergrid experiments:
- The experimental setting allows only for actions that increase coordinates at each time step. This is quite a strong condition and I wonder what are the effects of such condition. Specifically, It seems that the proposed approach is biased and it doesn’t satisfy general conditions, like detailed balance and ergodicity used in Markov chains (Note also that the proposed strategy is in effect a Markov chain). Consequently, the proposed strategy doesn’t provide any guarantee of correct sampling from the target. For instance, this is visible in Figure 3 (e.g. 2-D grid for 16x16, 60x60 and 16x16-hard and almost all cases with 4-D grid), where even the unbiased approach using the trajectory balance objective doesn’t converge to 0. Note that the phenomenon becomes even more pronounced with the increasing of dimensions. Can you comment on these aspects? Additionally, how does the proposed strategy perform by also allowing actions that decrease the coordinates?
- The baseline using the detailed balance objective is missing in Figure 3. Can you please include it in order to have a more complete picture of what is going on?
- Regarding the analysis of gradient bias. Why in Figure 5 (bottom) small-batch SubTB provides a better gradient estimate than small-batch TD? The explanation in the text seems overly stated (indeed the superiority is visible only at 500 training steps and in between 1500-2000 steps), can you be more precise? Additionally, in Figure 4 (right), SubTB is extremely biased compared to TB even for full batch size (K=10). Can you elaborate more on that?

About bit sequences experiments:
- Can you add the baseline using the detailed balance objective in Figure 7?


**Summary Of The Paper:**

The paper considers the recently developed GFlowNets, a family of models using a policy network to sequentially sample from an unnormalised target distribution defined on a discrete domain. Three main objectives are typically used to train such models, each enforcing a flow matching
(ensuring same incoming and outgoing flow at each node), a detailed balance (at each transition) and a trajectory balance condition (at global level). The work proposes a new objective unifying the last two conditions. The objective (i) extends the notion of balance to portions of trajectories, (ii) computes the balance contribution for each possible sub-trajectory and (iii) finally averages all these contributions. Importantly, the average used in the objective is governed by a single parameter, which is experimentally shown to control the bias-variance tradeoff. Indeed, a large parameter value offers similar behaviours to the trajectory balance objective (low bias, high variance), while a small parameter value shows similar effects to the ones obtained by the detailed balance objective (high bias, low variance). This bias-variance trade-off enables to reduce the time required to train GFLowNets and to generate better and more diverse solutions compared to the ones from previous objectives, as demonstrated by experiments on a hyper grid toy case, on small molecule synthesis and on three sequence generation tasks.

**Summary Of The Review:**

The paper proposes a new objective to train GFlowNets. The objective allows to tradeoff between existing local and global criteria (using the detailed and the trajectory balance conditions, respectively). Extensive experimental analysis provides support on the benefits of this unified view. The quality of the paper can be improved by including additional details about the training, including missing baselines and smoothen some strong statements in the explanations of the experimental analysis. Overall, the paper contributes to the advancement of GFlowNets from an empirical perspective.

---- POST REBUTTAL ------

First of all I would like to thank the authors for the clarifications to my questions. Also, I appreciated the inclusion of the baseline (based on detailed balance) in Figure 3, which allows for a better understanding.

Indeed, experiments in Figure 3 are the most important in the paper, as providing an empirical analysis of the bias-variance trade-off and giving insights about the behaviour of the strategy. From these figures I currently see three main potential issues:
1. While it is clear that SubTB($\lambda$) has the "lowest" curve, its convergence occurs at the same number of training iterations of the other approaches. Therefore, why is it possible to claim that the convergence is accelerated (as mentioned for example in the abstract)?
2. It is unclear why the baselines have strange behaviour. Indeed detailed balance is achieving both lower bias and lower variance than trajectory balance. This suggests that there is some additional factor influencing the results, probably the hyperparameters?
3. Hyperparameters seem to play a major role also for SubTB($\lambda$). Indeed it is unclear what is the effect of initialization (for instance note that in 16 x 16 hard the proposed strategy seems to start from a different condition from the other approaches) as well as how $\lambda$ is chosen.

In summary, the proposed objective unifies existing local and global objectives for GFlowNets and represents an interesting and novel idea. However, in absence of a theoretical analysis, the paper should provide a solid sensitivity analysis on the hyperparameters and utlimately a methodology for choosing them.

Based on these considerations and also based on others' reviews, I reduce my score.

---

> ### Author Response · Authors · 2022-11-09
> **Authors' response I: Answering questions**
>
> Thank you for your useful insights and suggestions about our paper. We begin by addressing your questions and concerns.
>
> > Some experiments (hypergrid, bit sequences) miss baselines.
>
> **Hypergrid:** Thank you for noting this omission. For a partial answer to your question, please see Figure A.2 in the Appendix, noting that $\lambda=0.01$ is nearly equivalent to DB and $\lambda=100$ to TB. We neglected to add DB to Figure 3 because TB was shown to outperform DB in prior work, but we will add it in a forthcoming update.
>
> **Sequences:** The Flow Matching is equivalent to a linear reparameterization of Detailed Balance when the state graph is a tree (as it is for autoregressive sequence generation tasks).  As such, the Detailed Balance baseline is already included in the sequence task results (in Figure 7 and Tables 2-3 "GFN - Flow Matching" is the same as "GFN - Detailed Balance").
>
> > Can you provide an algorithmic table to describe the training of GFlowNets and the computation of the proposed objective in order to make the paper self-contained?
>
> Thank you for this suggestion. We can consider adding an algorithmic box to the Appendix; for now, we provide a summary of the procedure described in Section 2.3:
>
> ```
> 1: Repeat:
> 2:   Sample trajectory tau from policy pi_theta
> 3:   Compute SubTB(tau) by equation (11)
> 4:   theta <- theta - eta d/dtheta SubTB(tau)
> 5. Until some convergence condition.
> ```
>
> > No supporting theory is developed to analyze the bias-variance tradeoff.
>
> Please see the response to all reviewers.
>
> > Is the partial trajectory objective computed at the end of or during each episode in a TD-like fashion?
>
> The SubTB objective is computed at the end of the episode, although other variants are possible. This only seems like a minor variation, but we leave it for future work and extensions.
>
> > Regarding the analysis of gradient bias. Why in Figure 5 (bottom) small-batch SubTB provides a better gradient estimate than small-batch TD? The explanation in the text seems overly stated (indeed the superiority is visible only at 500 training steps and in between 1500-2000 steps), can you be more precise? Additionally, in Figure 4 (right), SubTB is extremely biased compared to TB even for full batch size (K=10). Can you elaborate more on that?
>
> As you mention, in Figure 5 (bottom) SubTB only has a superior gradient estimate for a select set of training iterations.  In this plot we use the same sub-batch size, 64, as was used for training the GFlowNet.  However, for smaller sub-batches of size less than 10 the superiority of SubTB can be seen at nearly every iteration across training (see, e.g., the right side of Figure 4 at small values of K).
>
> With regard to Figure 4 (right), we note that this figure shows the similarity between small-batch DB, SubTB and TB gradients and the large batch TB gradient. Thus, we should only see an improvement in cosine similarity for **small** values of K, as demonstrated in the figure.  Figure 4 (left) shows the **self**-similarity between small and large batch gradients for each objective. There, as expected, the curve of SubTB lies above that of TB, indicating lower gradient variance.

---

> ### Author Response · Authors · 2022-11-09
> **Authors' response II: Correcting some possible misunderstandings**
>
> In this comment, we address some possible misunderstandings about GFlowNets in general and our paper in particular.
>
> > The sampling strategy doesn’t come with statistical guarantees on the quality of the obtained samples (like MCMC does). However, this is an intrinsic limitation of GFlowNets rather than of the proposed strategy.
>
> It is true that one can easily estimate the variance of *Monte-Carlo averages*, and one can assess the extent to which an MCMC trajectory seems to converge, but as far as we understand, there is no way to quantitatively evaluate the effect of missing modes in a given set of MCMC trajectories. Without extrinsic knowledge: if these modes are exponentially difficult to find by the MCMC random walks, then there will be no signal to alert us to that effect.
>
> The strategy of GFlowNets to obtain samples is very different from MCMC in that GFlowNets use a generative ML approach to discover potential patterns among the samples (and their energy or reward) making it possible to guess (i.e., generalize to) yet unvisited modes. Yes, if the learner does not discover those modes, we still cannot know we are missing them (short of using an exponentially expensive calculation or knowing them ahead of time). Note that this is similar to the general problem in ML of knowing how far a learner's function is from the Bayes-optimal function. In general there is no way to know. However, what we can do with GFlowNets is use the GFlowNet's probability estimates of the generated samples and measure their correlations with the reward function.
>
> > The experimental setting allows only for actions that increase coordinates at each time step. This is quite a strong condition and I wonder what are the effects of such condition.
>
> The action space for the grid environment is taken from previous works on GFlowNets, where decrementing coordinates was not permitted. In past work, this choice was made to ensure acyclicity of the state graph. Our grid environment follows past work, but additionally tests larger grids and sparser reward functions where the contrast between SubTB and previously proposed GFlowNet training objectives becomes more visible.
>
> > It seems that the proposed approach is biased and it doesn’t satisfy general conditions, like detailed balance and ergodicity used in Markov chains.
>
> The DAGness property assumed in the published GFlowNet theory works does not introduce any bias, since the generative policy can still reach any feasible terminal state, i.e., potentially generate any object in the domain of interest. GFlowNets should really be thought as inference machines or generative models rather than a general-purpose RL method. Standard RL could be used to generate objects sequentially as well, but with a different objective (maximizing the reward rather than sampling). That being said, the paper [Bengio et al., 2021b] shows how one can incorporate actions that allow changing coordinates in any direction if one adds an extra variable to the state, such as a time stamp, which always increases after each action, thus guaranteeing that the set of trajectories still forms a DAG. (It can also be shown that much of the GFlowNet theory continues to hold even if the state graph contains cycles, under mild assumptions such as nonvanishing of the policy, but this is beside the point of this paper.)
>
> Concerning detailed balance, this is a condition for MCMC methods and does not strictly make sense in the GFlowNet setting, since GFlowNet samplers do not sample a recurrent Markov process and instead generate the object in a single shot. However, there is an interesting mathematical analogy between the MCMC detailed balance condition and the GFlowNet constraint of the same name: the random process defined by the forward policy in a GFlowNet can be seen as a transient Markov chain, where the terminal states are absorbing; the backward policy is its reverse.
>
> > Consequently, the proposed strategy doesn’t provide any guarantee of correct sampling from the target. For instance, this is visible in Figure 3 (e.g. 2-D grid for 16x16, 60x60 and 16x16-hard and almost all ceased with 4-D grid), where even the unbiased approach using the trajectory balance objective doesn’t converge to 0.
>
> First, the plots for the hypergrid environment are showing not the objective, but the empirical L1 error between states sampled from the policy and the target distribution. In the limit of an infinite number of samples used to estimate L1, the curves in Figure 3 would approach 0. We use 200k samples; even for a perfect sampler, the expected L1 error would be positive.
>
> Second, the trajectory balance (and also the SubTB) objective **does** in fact converge to 0 in these environments. This **guarantees correct sampling from the target**, as is proved in different past work for different objectives ([Bengio et al., 2021a] for FM, [Bengio et al., 2021b] for DB, [Malkin et al., 2022] for TB).

---

> ### Author Response · Authors · 2022-11-17
> **Follow-up**
>
> We thank the reviewer for their time and efforts to read our work and provide comments and suggestions. Based on the feedback, we have addressed all of the concerns mentioned, a summary of which is as below:
>
> 1. **Algorithmic table to describe the training:**
> We have added this detail to our previous response and we agree this will help in making the paper self-contained.
>
> 2. **Questions about the hypergrid setting:**
> We have responded to this in our previous answer. We hope this helps in addressing the questions the reviewer had.
>
> 3. **Missing baseline of Detailed Balance in Fig 3:**
> We have added this baseline and updated it in our submission. Thank you for pointing this out.
>
> 4. **Analysis of gradient bias:**
> We provide a discussion on this in our previous comment. We would be happy to discuss this more if the reviewer still has any more questions or concerns.
>
> 5. **Detailed balance objective in Figure 7:**
> We specify how this baseline is already included in our results in Fig. 7 in our previous response. We hope this helps in completing the picture.
>
> Other questions related to sampling and convergence are also addressed in our previous response.
>
> We are happy to provide more details and extensions for any of the experiments and discussions in our paper. Please let us know if there are any other questions or concerns. We thank you in advance.

---

> > ### Comment · Reviewer_dWW3 · 2022-11-26
> > **Thank you**
> >
> > Dear authors,
> >
> > thank you for the answers to my questions. I think that the paper has improved in terms of quality/clarity over the pre-discussion period.
> > However, I think that some additional work is required to strengthen the proposed solution in absence of a theory on the bias-variance trade-off. For further details, please refer to the updated comments in the summary of the review.
> >
> > Best

---

> > > ### Author Response · Authors · 2022-12-04
> > > **Response to updated summary**
> > >
> > > We would like to thank the reviewer for taking a look at the updated submission and to clarify a few things based on the post-rebuttal comments:
> > >
> > > 1. SubTB($\lambda$) finds more modes quicker, and also has the lower L1 curve, as shown in Figs. 2 and 3, which is evidence of faster convergence.
> > >
> > > 2. Our analysis and claims about gradient variance are about **stochastic gradients** (see Section 4.1.1, Fig. 4, and Fig. 5), not about the variance between different **random seeds** (shaded regions in Fig. 3). We kindly ask you to refer to Section 4.1.1, which provides an in-depth analysis of the behavior of the gradients for each method, supporting the hypotheses about the benefits of SubTB. In addition, please refer to our response (Authors' response to all reviewers: On theoretical analysis of SubTB), which explains the difficulty of building a theoretical analysis. Our empirical analysis and ablations, including the impact of gradient variance, aim to compensate for this.
> > >
> > > 3. We would like to mention again that for the proposed SubTB($\lambda$) objective and for all other methods, we provide full details on our choices of hyperparameters, including lambda and learning rates for each experiment, in the Appendix. As mentioned in the Appendix, we try a wide range of hyperparameter settings for all methods and pick the one which performs the best. Other than this, all methods are run on exactly the same environments and conditions. On the 16x16 grid, all methods (including the proposed method) start with identically distributed random initializations, and the mentioned difference could be because the L1 is first computed **after** a few steps of gradient descent have been made.

---

### Official Review · Reviewer_VBco · 2022-10-25

**Confidence:** 2
**Clarity, Quality, Novelty And Reproducibility:** The paper is novelty, but the motivat…
**Correctness:** 3
**Technical Novelty And Significance:** 3
**Empirical Novelty And Significance:** 2
**Recommendation:** 5

**Strength And Weaknesses:**

The proposed method is novel, and the paper is well-written. However, I have some concerns and thus do not recommend acceptance.

Q1. My biggest concern is that the experiments are quite simple. How well does $SubTB(\lambda)$ perform over complex tasks in high-dimensional environments?

Q2. The motivation of $SubTB(\lambda)$ is not very celar. More theoretical analysis is required to explain why $SubTB(\lambda)$ improves the convergence of GFlowNets.

Q3. The current version lacks discussion about the limitations of $SubTB(\lambda)$.


**Summary Of The Paper:**

The paper propose a novel GFlowNet training objective called subtrajectory balance $SubTB(\lambda)$ that can learn from partial The paper proposes a novel GFlowNet training objective called subtrajectory balance $SubTB(\lambda)$ that can learn from partial action subsequences of varying lengths. Empirical results demonstrate that $SubTB(\lambda)$ can improve the convergence of GFlowNets in some simple environments.

**Summary Of The Review:**

I have some concerns and thus do not recommend acceptance.

---

> ### Author Response · Authors · 2022-11-09
> **Authors' response**
>
> We would like to thank the reviewer for the thoughtful comments and feedback on the paper.
>
> > My biggest concern is that the experiments are quite simple. How well does SubTB($\lambda$) perform over complex tasks in high-dimensional environments?
>
> We would like to clarify that we already consider several complex high-dimensional tasks. While "high-dimensional" has no precise meaning in discrete spaces, we experiment with tasks that are complex in various ways:
> - **Large state spaces**: Specifically, in Section 4.2 (small molecule synthesis), the size of the state space is on the order of $10^{12}$. For the antimicrobial peptide generation experiments in Section 4.3.2, the size of the state space is $21^{60}$, and for the fluorescent protein generation experiments in Section 4.3.3 it is $20^{237}$.
> - **Large action spaces**: The experiments in Sections 4.2 and Section 4.3.1 also feature large action spaces. For example, in the bit sequence experiments, the size of the action space is as large as 1024 (when k=10).
> - **Long action sequences**: For the experiments in Section 4.1, we consider much larger grids than used in the previous works (as well as a harder reward configuration, which we call the "hard" grid). The experiments in Section 4.3.3 also feature very long (length 237) action sequences.
>
> We emphasize that these tasks, which are practically relevant and have impactful downstream applications, have been shown to be incredibly challenging for existing methods in prior work.
>
> > The motivation of SubTB($\lambda$) is not very clear.
>
> The main motivation of SubTB($\lambda$) is to bridge a gap between the DB loss (which is slow because of poor credit assignment) and the TB loss (which has high gradient variance). SubTB has lower variance than TB while still inducing faster convergence. We discuss this motivation in the abstract, the introduction, and the end of Section 2.3 ("Hypothesized benefits").
>
> A secondary motivation is the ability to train GFlowNets on partial episodes of any length, which is briefly discussed in Appendix A.1 and could be important in many applications, such as those related to RL environments.
>
> > More theoretical analysis is required to explain why SubTB($\lambda$) improves the convergence of GFlowNets.
>
> Please see the answer to all reviewers about theoretical analysis.
>
> > The current version lacks discussion about the limitations of SubTB($\lambda$).
>
> Thank you for bringing up this point. One clear limitation of SubTB($\lambda$) is related to the hyperparameter $\lambda$. In the current work, $\lambda$ is tuned manually. It could be hyper-optimized as is done with other hyperparameters, but the framework can be extended to learn this as well and we leave such an extension for future work.

---

> ### Author Response · Authors · 2022-11-17
> **Follow-up**
>
> We sincerely thank the reviewer for their constructive feedback and suggestions. We have addressed the main concerns mentioned in the review:
>
> 1. **Performance over complex tasks and high-dimensional environments:**
> In our previous response, we have added more details on the complexity of the environments we have considered in this work and have specified how each of these environment is complex and high-dimensional. In fact, the benefits of our proposed algorithm are more visible complexity of the environment increases. In addition, we would like to mention that in our submission, have also added more complex variants of the previously studied environments, such as the harder reward version of the hyper grid environment (Section 4.1) as well as two new datasets, fluorescent protein generation (Section 4.4.3) and inverse protein folding (Appendix F). The proposed method outperforms all previous GFlowNet training objectives in all of these environments in terms of both faster convergence and lower variance.
>
> 2. **More clarity on motivation and theoretical perspective:**
> We provide more pointers to the motivation discussed throughout the paper in our previous response. We also provide more details on considering the theoretical aspects of our method, and detail out why we chose an empirical approach. Moreover, we provide a thorough analysis of many different scenarios through our empirical experimentation, including study of the behaviour of gradients for each method (Sec 4.1.1) as well as different training scenarios that can help with exploration exploration (Fig. A.3). We also empirically show how the proposed method provides a smooth transition between the previously proposed methods of TB and DB (Fig. A.2) which helps broaden the understanding of the different GFlowNet training objectives that can be considered the extreme cases of the more general form of the proposed method, SubTB($\lambda$) through the hyperparameter $\lambda$.
>
> 3. **Discussion about the limitations of the current method:**
> We mention some of the limitations of the current method, such as the tuning of the lambda parameter. We believe that studying ways to address these limitations would be an interesting subject for future work.
>
> 4. **Novelty and contribution:**
> The proposed method provides a general framework that can be used to improve the training of GFlowNets. It outperforms all previous methods, and we find better convergence and variance properties on both existing and new as well as more difficult versions of the environments. In addition, the proposed method provides a general framework to think about the different GFlowNet objectives such that the previously proposed objectives of Trajectory Balance and Detailed Balance can be seen as special versions of the proposed method, SubTB($\lambda$) through the parameter $\lambda$. This also helps in extending the framework of GFlowNets to learning from partial trajectories of any length, which was not possible in the previous methods. In addition, we can see better training properties of GFlowNets on much more sparser and complex reward distributions, which was not possible before. The proposed method thus opens gates to richer environments and training scenarios for GFlowNets.
>
> We are happy to work to improve to improve our submission further and would greatly appreciate any more comments, suggestions, or feedback from the reviewer. Thank you again for your time and helpful review.

---

### Author Response · Authors · 2022-11-09
**Authors' response to all reviewers: On theoretical analysis of SubTB**

This comment addresses a common concern raised by all reviewers:

> Reviewer dWW3: No supporting theory is developed to analyze the bias-variance tradeoff.
>
> Reviewer VBco: More theoretical analysis is required to explain why SubTB($\lambda$) improves the convergence of GFlowNets.
>
> Reviewer aGGf: In the paper, the authors show how it provides better experimental results but don't argue how it advances theoretical understanding of GFlowNets.

This is true; we opted for an empirical analysis of gradients to explain and illustrate the hypothesized benefits of SubTB. However, let us try to explain the challenges in developing theory to explain the bias-variance tradeoff that led us to change our focus to experimental evidence of variance reduction.

As we discuss in Section 2.3, the state flow at the end state $s$ of a subtrajectory estimates the expected "tail" of the the TB objective for trajectories that pass through $s$. This leads to the intuition that SubTB and DB have lower variance, since the regression objective for policy logits and state flows preceding $s$ is deterministic, while for TB it is stochastic (as it depends on the policies and rewards that are accumulated *after* $s$ in a trajectory). However, it is possible to construct counterexamples -- particular values of the state flows and policies at which SubTB in fact has *higher* variance, even if such values are rarely attained during training.

We could hope for a weaker result by making additional assumptions. For example, we can take the state graph to be a tree (so the backward policy is trivial) or assume that the state flow function is perfectly optimized and analyze only the gradients of policy logits (as we do empirically in Appendix A.2). However, even in this setting examples exist in which DB and SubTB have higher variance than TB. One important source of the difficulty is that the loss for a trajectory that includes a given edge $s\rightarrow t$ sends a gradient not only to the policy $\log P_F(t|s)$, but also to all sibling edges, $\log P_F(s'|s)$ for children $s'$ of $s$.

Finally, variance alone does not guarantee faster convergence in the presence of (a) a nonstationary objective (trajectories are sampled on-policy or using a tempered policy) and (b) the use of momentum-enhanced optimizers (we are not aware of existing theoretical analyses of Adam in RL settings). **The main contribution of our paper lies in the superior performance of SubTB on several difficult tasks; variance reduction is an explanation for this result for which we provide some empirical evidence.**

---

### Decision · Program_Chairs · 2023-01-20

**Decision:**

Reject

**Justification For Why Not Higher Score:**

Lack of rigor in empirical protocols as it relates to choosing of hyperparameters and testing of ablations suggested by the reviewers. Given that the work is primarily of an empirical nature, this seems critical for acceptance.

**Justification For Why Not Lower Score:**

N/A

**Metareview: Summary, Strengths And Weaknesses:**

The paper proposes a new algorithm for resolving the bias-variance tradeoff in training of GFlowNets. The algorithm is inspired from TD(\lambda) in the RL literature and proposes to train the model over partially incomplete training trajectories. Empirical results demonstrate the strength of the approach on environments used in prior literature on GFlowNets.

On the positive side, most of the reviewers appreciated the idea, the writing of the paper, and some of the empirical results. On the negative side, there were a set of overlapping concerns with the execution and the claims of the work. The work builds on a recent body work that lacks theoretical results (GFlowNets) --- which is fine in my opinion as long as there are empirical benefits --- but there is an extra onus on the authors to set a high bar for empirical rigor. This goes beyond listing hyperparameter selection protocols and should include justifications grounded in practice, such as some reviewer suggestions on doing a sensitivity analysis. A few concerns from the reviewers that could have been addressed better with empirical evidence:
- Choosing best hyperparameters from a grid search can leak information; is there an automated procedure e.g., held-out performance? if not, is there a sensitivity analysis to justify the robustness of different hyperparameter choices?
- One of the reviewers highlighted that the different methods do not even start from the same initial performance, which is  indeed a bit concerning. Here, the authors claim it *could be* because they do not plot the L1 at iteration 0 --- evidence of the corrected numbers would have been preferable as opposed to speculating about a potential bug.
- Another reviewer was curious about the potential use of averaging of sequences for variance reduction (eg, using replay buffers), which is a reasonable question that merits a careful discussion/ablation to better understand the gains of the proposed method.

Overall, I encourage the authors to view some of the reviewer suggestions constructively to improve the empirical rigor of the work for a future submission.